# Interface-type tunable oxygen ion dynamics for physical reservoir computing

Zhuohui Liu[1,2,9], Qinghua Zhang[1,3,9], Donggang Xie[1,8,9], Mingzhen Zhang[1,4], Xinyan Li [1,2], Hai Zhong[1,5], Ge Li[1,4], Meng He[1], Dashan Shang [6], Can Wang [1,4], Lin Gu[7], Guozhen Yang[1], Kuijuan Jin [1,4] ✉ & Chen Ge [1,4] ✉

Reservoir computing can more efficiently be used to solve time-dependent tasks than conventional feedforward network owing to various advantages, such as easy training and low hardware overhead. Physical reservoirs that contain intrinsic nonlinear dynamic processes could serve as next-generation dynamic computing systems. High-efficiency reservoir systems require non-linear and dynamic responses to distinguish time-series input data. Herein, an interface-type dynamic transistor gated by an $Hf_{0.5}Zr_{0.5}O_2$ (HZO) film was introduced to perform reservoir computing. The channel conductance of Mott material $La_{0.67}Sr_{0.33}MnO_3$ (LSMO) can effectively be modulated by taking advantage of the unique coupled property of the polarization process and oxygen migration in hafnium-based ferroelectrics. The large positive value of the oxygen vacancy formation energy and negative value of the oxygen affinity energy resulted in the spontaneous migration of accumulated oxygen ions in the HZO films to the channel, leading to the dynamic relaxation process. The modulation of the channel conductance was found to be closely related to the current state, identified as the origin of the nonlinear response. In the time series recognition and prediction tasks, the proposed reservoir system showed an extremely low decision-making error. This work provides a promising pathway for exploiting dynamic ion systems for high-performance neural network devices.

Brain-inspired neuromorphic computing architectures have attracted increasing research and development interest in highly efficient computing with increasingly complex data processing tasks[1–4]. Among these, a series of recent novel electronic devices imitating many functions of synapses or neurons have been developed. Various tasks, such as pattern recognition[2,5,6], signature confirmation[7],

and auto-drive[8] can be achieved through the construction of neural networks. The formerly mentioned applications are mainly based on the feedforward network of an artificial neural network (ANN), which is suitable for executing static tasks such as pattern recognition or object detection[9]. However, the analysis and prediction of temporal tasks are still difficult in this framework. By contrast, the recurrent

[1]Beijing National Laboratory for Condensed Matter Physics, Institute of Physics, Chinese Academy of Sciences, 100190 Beijing, China. [2]College of Materials Science and Opto-Electronic Technology, University of Chinese Academy of Sciences, 100049 Beijing, China. [3]Yangtze River Delta Physics Research Center Co. Ltd., 213300 Liyang, China. [4]School of Physical Sciences, University of Chinese Academy of Science, 100049 Beijing, China. [5]School of Physics and Optoelectronics Engineering, Ludong University, 264025 Yantai, Shandong, China. [6]Key Laboratory of Microelectronic Devices and Integrated Technology, Institute of Microelectronics, Chinese Academy of Sciences, 100029 Beijing, China. [7]Beijing National Center for Electron Microscopy and Laboratory of Advanced Materials, Department of Materials Science and Engineering, Tsinghua University, 100084 Beijing, China. [8]Present address: School of Physical Sciences, University of Chinese Academy of Science, 100049 Beijing, China. [9]These authors contributed equally: Zhuohui Liu, Qinghua Zhang, Donggang Xie. ✉ e-mail: kjjin@iphy.ac.cn; gechen@iphy.ac.cn

neural network (RNN)[10] can be used as a promising candidate for dynamic data processing, such as voice, video, and chaotic systems[11]. As a variation of RNN, reservoir computing can avoid training the weight of the reservoir and settles based on a few global parameters related to input information. Hence, the complexity can dramatically be decreased without tracking each particle of the system. In addition, less computing costs and time may be obtained since only the output network is trained. Reservoir computing is a dynamic system that is able to map the input into a high-dimensional space through the short-term memory property[12]. In particular, a nonlinear mapping procedure can transform complex input into linear discernible states of the reservoir system[13]. The resulting output will then be trained through a single-layer network. Initially, software-level reservoir computing based on existing CMOS platforms was used to perform relevant dynamic tasks[14]. However, CMOS devices do not have intrinsic characteristics of dynamic response, so the processing of nonlinear dynamic tasks requires the combination of complex algorithms and large-scale integrated devices, which leads to unnecessary hardware consumption[15]. Fortunately, this problem can be resolved through the implementation of physical reservoir computing (PRC) with spontaneous nonlinear processes[15]. Due to the highly adaptive and flexible dynamic characteristics in the physical systems, more reliable PRC hardware with smaller size is expected to further enhance the performance of dynamic processing[16]. In summary, the implementation of PRC requires some particular features[13]. First, the states of reservoirs depend not only on the current input but also on recent inputs, thereby requiring short-term memory for efficient reservoir computing. Second, the system needs to generate distinguishable states responding to different inputs, thereby requiring highly nonlinear response devices[17]. In recent decades, many studies have demonstrated the implementation of PRC systems with applications in speech recognition[18], chaotic prediction[19,20], electric consumption prediction[21], and fingerprint identification[16] based on two-terminal memristors[18,22,23], spintronic oscillators[24], programmable logic gate arrays[25], photonic module devices[26–28], and quantum devices[29].

Three-terminal PRC reservoir systems have attracted much attention since larger design freedom can be provided through the multiterminal structure of transistors[30]. Besides, the separated read and write terminals help stabilize the electrical processes, thereby avoiding huge read currents and effects caused by Joule heating[31]. Currently, the dynamic processes of lithium ions[32], hydrogen ions[33], organic ions[34], and ferroelectric polarization[30,35] are used for reservoir computing. Among all ion-gating transistors, considerable potential has been paid to oxygen ion dynamic-based reservoirs since these can also be feasible for conventional lithography techniques and stable under various temperatures. Furthermore, oxygen ion dynamic-based reservoirs can be fully compatible with complementary metal-oxide-semiconductor (CMOS) technology and are environmentally friendly compared with lithium and hydrogen ions[36]. Nevertheless, transistors induced by interface oxygen ion dynamics have not yet been introduced to reservoir systems.

The discovery of hafnium-based fluorite ferroelectrics broke the bottleneck constraints for realizing ultrathin ferroelectric films highly compatible with existing CMOS processes[37,38]. Hafnium-based fluorite ferroelectric materials are advantageous in terms of high-speed, low-power neuromorphic computing, attracting interest from materials science to engineering fields[39]. In particular, it has been reported that the polarization process of $Hf_{0.5}Zr_{0.5}O_2$ (HZO) is strongly coupled with the oxygen migration at the heterojunction[40]. As a result, the oxygen ions in the $La_{0.67}Sr_{0.33}MnO_3$ (LSMO$_3$) buffer layer can be extracted, and even phase transition from perovskite (LSMO$_3$) to brownmillerite (LSMO$_{2.5}$) will be observed[40]. In other words, HZO can act as a good conductor of oxygen ions[41,42]. Meanwhile, the extraction and insertion of oxygen ions affect the transport properties in LSMO films by changing the oxygen stoichiometry and valence state of metal cations in films[43]. Although LSMO can hardly be modulated as a Mott oxide[44–46], obvious changes in conductance can be realized via the coupled property of polarization switching and ion migration in HZO. The stability for offsets of oxygen stoichiometry depends on energy preference, leading to dynamic relaxation when oxygen vacancies cause unfavorable energy states[47].

Herein, an interface-type tunable dynamic system with a ferroelectric HZO gate and LSMO as the channel layer was developed and tested. The interaction between ferroelectric polarization and migration of oxygen was employed to manipulate the channel conductance. The capture and release of oxygen ions can effectively modify the conductance in the LSMO channel film, and the transfer curve shows a 500% current variation. The existence of oxygen vacancies was confirmed through transmission electron microscopy (TEM) equipped with electron energy loss spectrometry (EELS). Under an applied positive pulse to diminish the conductance, obvious nonlinear and volatile relaxation appeared, which can be described by a double exponential function. Taking advantage of such nonlinear and volatile processes, reservoir computing applications were conducted for tasks such as static pattern recognition, voice recognition, waveform classification, and chaotic prediction. Overall, novel insights into the development of physical high-precision reservoir computing were provided which are useful for future applications.

## Results

### PRC system with an ion dynamic transistor

Reservoir computing is an extended framework of neural network. A conceptual illustration of the dynamic system is provided in Fig. 1a. The reservoir system was based on a typical ferroelectric transistor device with a Mott oxide channel. A schematic structure of the ferroelectric transistor is displayed in Fig. 1b. Here, the conductance of the LSMO channel layer was measured between the source and drain electrodes. The reversible structural transformation between the pristine and oxygen-deficient phases in the channel layer is schematically depicted in Fig. 1c. The HZO film was then deposited as the ferroelectric gate. Both the source-drain and gate electrodes were made of platinum films with thicknesses of 40 nm and 20 nm, respectively. More details about device fabrication can be found in the "Methods" section.

X-ray diffraction (XRD) of the HZO film showed a mainly orthorhombic phase with <111>-oriented plane[37,48–51], which is useful for ensuring good ferroelectricity of films (Supplementary Fig. 1a). To confirm the polarization properties, an HZO/LSMO heterostructure was prepared on a $(LaAlO_3)_{0.3}$-$(SrAl_{0.5}Ta_{0.5}O_3)_{0.7}$ (LSAT) substrate. The polarization-voltage (P-V) hysteresis and piezo-response force microscopy (PFM) measurements were then conducted. As shown in Supplementary Fig. 1b, partial polarization can be obtained under different voltages with a remanent polarization ($P_r$) value of approximately 26 μC/cm². Moreover, 180° phase reversal in the phase image and local hysteresis loop also indicated ferroelectric switching in the film[52] (Supplementary Fig. 2b–d).

Figure 1d shows the current variation under different positive gate voltages. Here, the amplitude values of 2, 2.5, and 3 V with a width of 20 s and an interval of 100 s were used. Obviously, the application of positive voltages decreased the channel conductance. As the voltage rose, the modulation range was expanded simultaneously. Under an applied 3 V pulse, the dynamic range of the drain current was approximately 30 nA. In addition, the transfer curve test revealed a dynamic range of approximately 500% (Supplementary Fig. 3a). However, the channel current relaxes toward the initial state after the removal of the pulse. By contrast, the application of negative pulses enhanced the channel current, and the final state was maintained steadily for a long time (Supplementary Fig. 4). Thus, the modulation of channel conductance by the HZO gate layer was asymmetric. Under an applied positive gate voltage, obvious volatile variation was

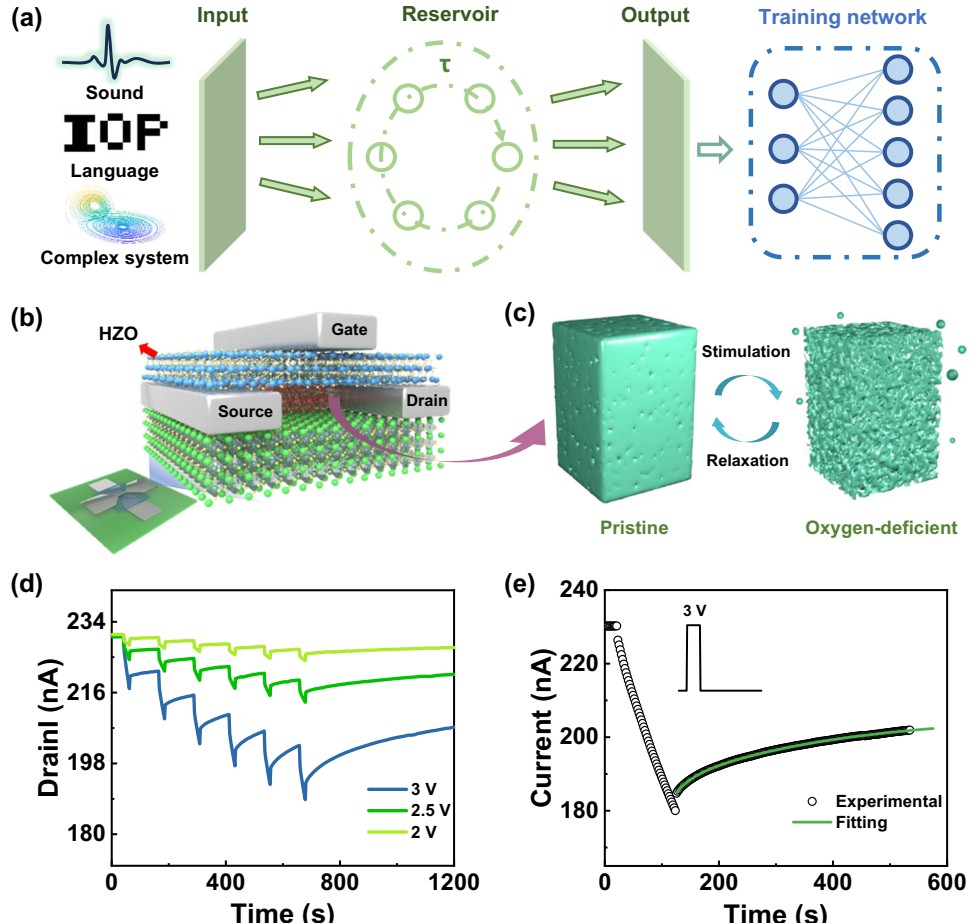

**Fig. 1 | Ferroelectric Hf$_{0.5}$Zr$_{0.5}$O$_2$-based FET with asymmetry modulation process for reservoir computing. a** Schematic representation of the dynamic reservoir system. The input is fed into a reservoir composed of a series of virtual nodes with a fixed time step. The output vector is a linear combination of the values generated by the reservoir, and weights are trained through linear regression. **b** Device structure. **c** Schematic representation of the structural transition between the pristine and oxygen-deficient phases. **d** Evolution of the drain current as a function of time during different positive voltage stimulations. **e** The nonlinear fitting (green curve) of the relaxation process after removal of the positive voltage pulse in comparison with the experimental results (hollow circle).

obtained. The decay in current was analyzed and the results fit a double-exponential function well (Fig. 1e), as shown by Eq. (1):

$$I = A_1 e^{\frac{-t}{\tau_1}} + A_2 e^{\frac{-t}{\tau_2}} + I_{\infty} \tag{1}$$

where $\tau_1$ and $\tau_2$ represent two characteristic time constants with values of 26.9 s and 275.9 s, corresponding to the fast and slow decay processes, respectively. $A_1$ and $A_2$ are constants. $I_{\infty}$ is a current constant of the relaxation process.

Various pulses with different durations and numbers of pulses were also applied for both positive and negative stimulations (Supplementary Fig. 5). The variation in the current range expanded as a function of the number and width of pulses. For Mott transistors, many efforts have previously been made to enlarge the regulation range through the use of materials with lower carrier density[53,54], modification through thickness[55,56], and strain engineering[57]. However, expanding the regulation range in Mott transistors is still challenging[58]. The proposed device provided an effective way to manipulate the conductance state in a strongly correlated material. As depicted in Supplementary Fig. 5, similar asymmetry modulation was observed in the opposite direction, indicating different dynamic processes between the two procedures.

## Mechanism of dynamic transistors for reservoir computing

To explore the physical mechanism of the asymmetric conductance change, we carried out high-resolution transmission electron

microscopy (TEM) measurements in HZO/LSMO epitaxial heterostructures on LSAT substrates. The characterization results revealed a high crystallinity and an atomically sharp interface between HZO and LSMO. Ex situ observations of the atomic lattice in the heterostructures before and after the application of pulses are depicted in Fig. 2a–c, respectively. The amplitude of the voltage pulse is ±3 V. Here, the defined "Negatively Pulsed" sample is first pulsed by a positive voltage and then pulsed by the negative voltage. The amplified images of enclosed regions with a square in Fig. 2a–c are provided in Fig. 2d–f, respectively. The atomic observations illustrated that LSMO maintained the perovskite structure after the application of positive pulses. However, the average out-of-plane lattice parameter of pulsed LSMO was slightly enhanced compared to that of the pristine film (Supplementary Fig. 6), indicating a trend of phase transition from perovskite to the oxygen-deficient structure. However, the structural transition from perovskite to brownmillerite structures was absent, possibly ascribed to the relaxation process during the ex situ measurements. The lattice parameters of the negatively pulsed sample were also calculated, and the decreased value of the out-of-plane lattice parameters suggests that the lattice expansion caused by oxygen vacancy has been restored. The phase transition should not only change the lattice parameters but also the valence of metal ions and bonding connection with oxygen ions.

To confirm the formation of oxygen vacancies and related effects, electron energy loss spectroscopy (EELS) was used to compare the

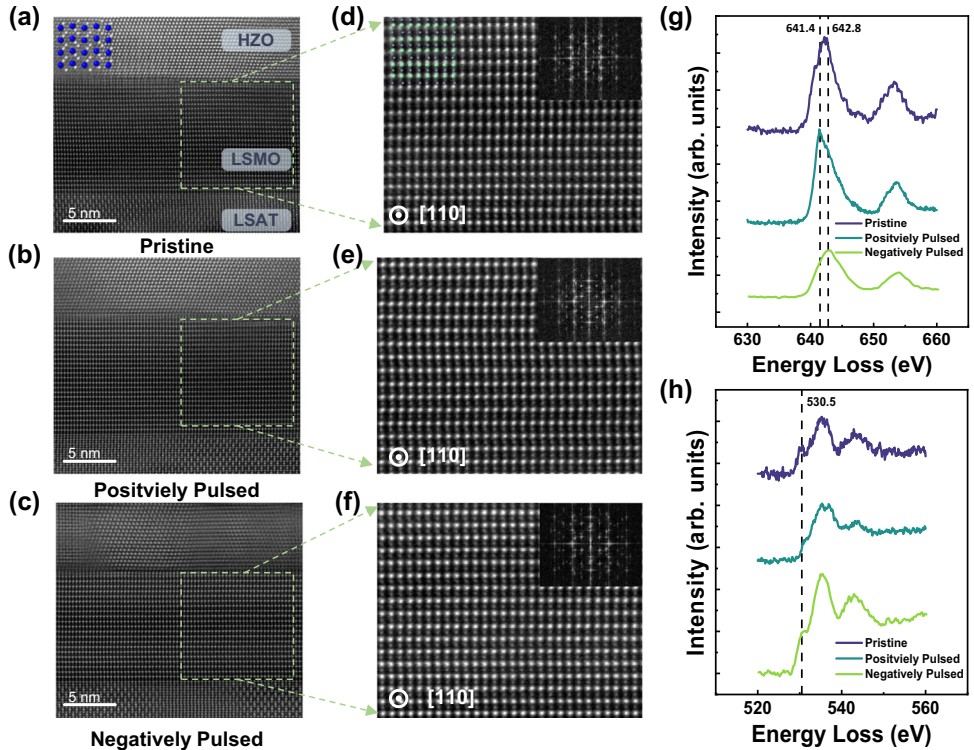

**Fig. 2 | Structural characteristics of the LSMO channel before and after pulse stimulation. a**–**c** STEM image of the **a** pristine, **b** positively pulsed and **c** negatively pulsed heterostructure. **d**–**f** Enlarged views of the selected areas of the boxes in (**a**–**c**). **g**, **h** STEM-EELS spectra of **g** Mn-$L_2$ and Mn-$L_3$ edges and **h** O-$K$ edges for the pristine and pulsed LSMO layers.

near-edge fine structure of the Mn-$L_2$/$L_3$ edge and O-$K$ edge in LSMO films on the LSAT substrate. Figure 2g compares the peaks of the Mn-$L$ edge before and after the application of electrical pulses. The as-grown LSMO was composed of $Mn^{3+}$ and $Mn^{4+}$. Additionally, the shift in the $L_3$ peak toward lower energy losses and the increased $L_3$/$L_2$ ratio indicate a reduction in the Mn oxidation state via the formation of oxygen vacancies[59–61]. Meanwhile, the comparison of the EELS spectra of the O-$K$ edge of the initial and pulsed LSMO films revealed a pre-peak at ~530 eV, caused by the hybridization between the O $2p$ and Mn $3d$ orbitals. The disappearance of pre-peak near 530 eV proved the existence of oxygen vacancies (Fig. 2h)[62]. Supplementary Fig. 7 shows the depth analysis of STEM-EELS spectra in pristine and pulsed LSMO films. The same spectral characteristics at different depths reveal that the modulation by oxygen ions can be effective throughout the film. Supplementary Fig. 8a, b summarizes the peak positions of the Mn-$L_3$ edge and the area ratio of the $L_3$/$L_2$ edge peaks for the pristine and pulsed films. The shift of the peak toward higher energy after negative voltage stimulation confirms the oxidation of $Mn^{3+}$. In addition, the transport properties of pristine and pulsed LSMO films in Supplementary Fig. 9 show obviously enhanced resistivity of LSMO after the application of positive gate voltages. Besides, the pulsed LSMO exhibited lower metal-insulator transition temperature ($T_p$) due to the suppression of the double exchange interaction through the creation of oxygen vacancies[61].

A schematic model of the proposed oxygen ion dynamic system is illustrated in Fig. 3. The migrations of oxygen ions and ferroelectric polarization during and after the electrical pulses were visible. Under applied positive pulses to the device, the oxygen ions interrelated with polarization were extracted from the LSMO layer, leading to the creation of oxygen vacancies in the film. The double exchange effect of charges in LSMO decreased, resulting in higher resistivity channels[61]. After the removal of pulses, the LSMO layer captured several oxygen ions from HZO, demonstrating good conduction of oxygen ions.

Meanwhile, oxygen ion filling resulted in the recovery of the resistivity of LSMO toward lower levels. Consistent with our measurement results, the obvious relaxing procedure was observed in Fig. 1d, e. In contrast, applying negative pulses drives oxygen ions to migrate to the LSMO layer. After the removal of negative pulses, the oxygen ions were still relatively stable since LSMO preferred an oxygen-rich state. As a result, the conductance state was identified as nonvolatile (Fig. 3e). According to computations based on the density function theory, the formation energy of oxygen vacancies in LSMO was identified as a relatively high positive value, meaning a stable ground state. In other words, LSMO was inclined to form an energy-favorable perovskite structure with fewer oxygen vacancies[47]. Besides, the affinity energy of LSMO was relatively low, benefiting the oxygen-rich structure. In particular, the LSAT substrate provided proper epitaxial conditions helpful for the reversibility of LSMO films[47]. Moreover, the modulation range of the channel through the HZO gate depended on the current structure of the LSMO layer. The pristine conductance state can be seen as a reflection of the LSMO lattice structure. As Fig. 1d shows, with degrading conductance, the relaxing procedure appeared more obvious because the degree of oxygen vacancies was initially high. It should be noted that the modulation of the channel conductance through ferroelectric polarization is typically nonvolatile due to its spontaneous polarization property[53,63]. However, our device shows an obvious volatile response under electrical gating, which indicates that the observed response characteristics are dominated by the migration of oxygen ions. In other words, the HZO film plays a more important role in this work as an oxygen ion conductor than as a ferroelectric gate.

The relation between the drain current and pulse voltage can be described by a piecewise function. When positive voltage pulses are applied to the gate, the degradation of the drain current is divided into two steps. In the beginning, only the extraction of oxygen ions exists, resulting in a sharp decreasing trend (Stage I, Eq. (2)). Afterward, the

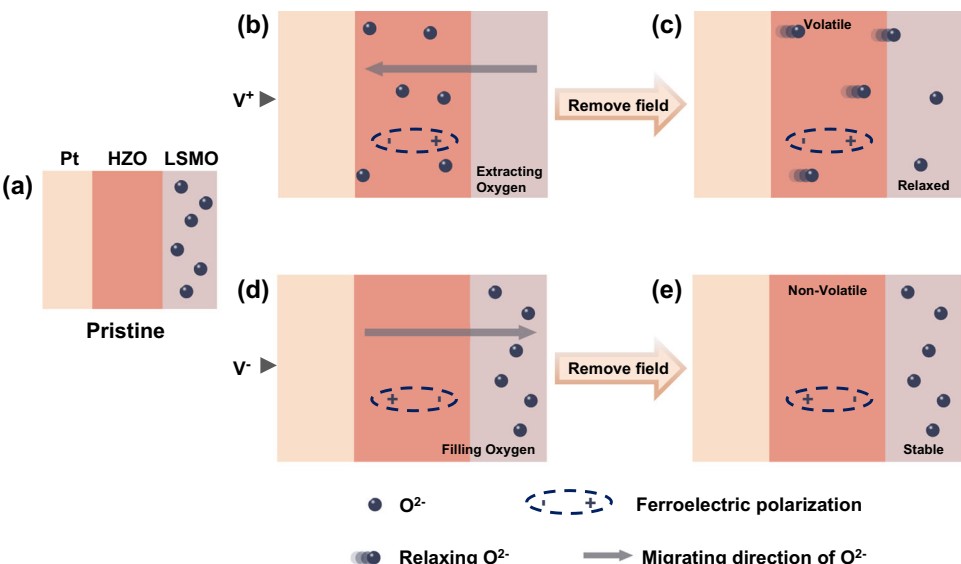

**Fig. 3 | Schematic illustration of volatile and nonvolatile regulation during the gating process in the proposed device. a** The initial state. **b** Under an applied positive pulse to the gate, in which the ferroelectric polarization is pointed upward, while the oxygen ions migrate away from the LSMO layer. **c** After the removal of the

voltage pulse, the oxygen ions spontaneously drafted to the LSMO layer. **d** During the application of a negative pulse, in which the polarization pointed in the downward direction, leading to the migration of oxygen ions back into the LSMO film. **e** After removing the negative pulse, the state of LSMO is kept nonvolatile.

spontaneous oxidization process would lead to competition between the separation and capture of oxygen ions due to the large negative value of the oxygen affinity of $LSMO_{3-x}$. As a result, the decreasing rate became slower. (Stage II, Eq. (3)). After the removal of the gating voltage, the relaxation of the drain current can also be considered as a two-stage process. At the moment of voltage removal (Stage III, Eq. (4)), fast relaxation was observed due to the relatively severe lack of oxygen ions in the LSMO channel. After capturing back an adequate number of ions into the channel, the relaxation process was slowed down (Stage IV, Eq. (5)). The dynamic process can be summarized as Eqs. (2)–(5), where $I_1$, $I_2$, $I_3$, and $I_4$ represent the current values under different states, $V$ is the applied gate voltage, and $I_0$ refers to the initial drain current. The coefficients A, k, b, and c are all constants. Among them, variable $t_{modu}$ is the duration of applying voltage, i.e., modulation time, and $t_{relax}$ is the relaxation time after removing the gate voltage. The detailed values of the parameters are provided in Supplementary Table 1. As depicted by the following functions, the evolution of the drain current depended on both the applied voltage and the former current state. More details about the fitting curve can be seen in Supplementary Fig. 10. This model can well describe the dynamic evolution of the device state, paving the way for later applications.

$$I_1 = k_1 V + k_2 V I_0 + I_0 \qquad (2)$$

$$I_2 = I_1 + (y_1 + A_1 e^{\frac{V}{b_1}}) \, t_{modu} \qquad (3)$$

$$I_3 = k_3 V + (y_2 + A_2 e^{-\frac{V}{b_2}}) I_2 + I_2 \qquad (4)$$

$$I_4 = I_3 + \frac{A_3}{1 + e^{\frac{I_3 - c_1}{b_3}}} \times \left(1 - e^{-\frac{t_{relax}}{c_2 + k_4 I_3}}\right) \qquad (5)$$

**Transistor-based reservoir system for pattern recognition**
The recognition process was tested starting with a simple task of recognizing letters from an input image. For example, the image letter "S" in Fig. 4a is composed of 20 pixels either black ("1") or white ("0"). This was then divided into 5 parallel rows, each containing 4

consecutive pixels, fed into a transistor reservoir as a 4-timestream input stream. For the letter "S", the time-series inputs [0111], [1000], [0110], [0001], and [1110] were obtained. More patterns of the letter are depicted in Supplementary Fig. 11, and the experimental evolution of the above input transformation process is shown in Fig. 4a. A timeframe containing a single write pulse (amplitude of 3.3 V and the pulse duration of 6 s) as "1" and no pulse representing "0" was presented. During all measurements, the read voltage applied to the source-drain was 0.1 V. After the application of all 4-timestream inputs to the reservoir, the information of an entire image was nonlinearly projected into the reservoir system. The volatile and nonlinear properties of the system induced different timeframe input-independent current changes. In Fig. 4b, the overall width and amplitude of the voltage were the same for inputs [0111] and [1110]. Additionally, the final outputs collected after the removal of all pulses for 10 s (shaded region in Fig. 4b) looked distinct, which represent the states of the reservoir.

During the classification, a single-layer fully connected neural network with a size of 5×20 was used to perform the letter recognition task. The Softmax function was selected as the activation function of the network, and the weights were updated based on the back-propagation algorithm. With 4-timestream pulses, a total of 16 different states were obtained, as shown in Fig. 4c, with each line representing an independent evolution of the reservoir state. The variation in drain current in Fig. 4d revealed separable outputs of the reservoir proving the nonlinearity and volatile properties of the device. After training the readout, the system can correctly classify digital letters with 100% accuracy within less than 50 training epochs. In order to verify the performance of the classification tasks is indeed due to the reservoir properties of the transistor, we performed the same tasks using a linear model without reservoir computing processing. The classification accuracy decreased to 80% (Supplementary Fig. 12). The above results verified the feasibility of static pattern recognition using our reservoir system.

**Transistor-based reservoir system for speech recognition**
The performance of the dynamic FET system on temporal classification tasks was further evaluated by spoken information recognition. The input data consisted of audio waveforms from our colleges. The goal of

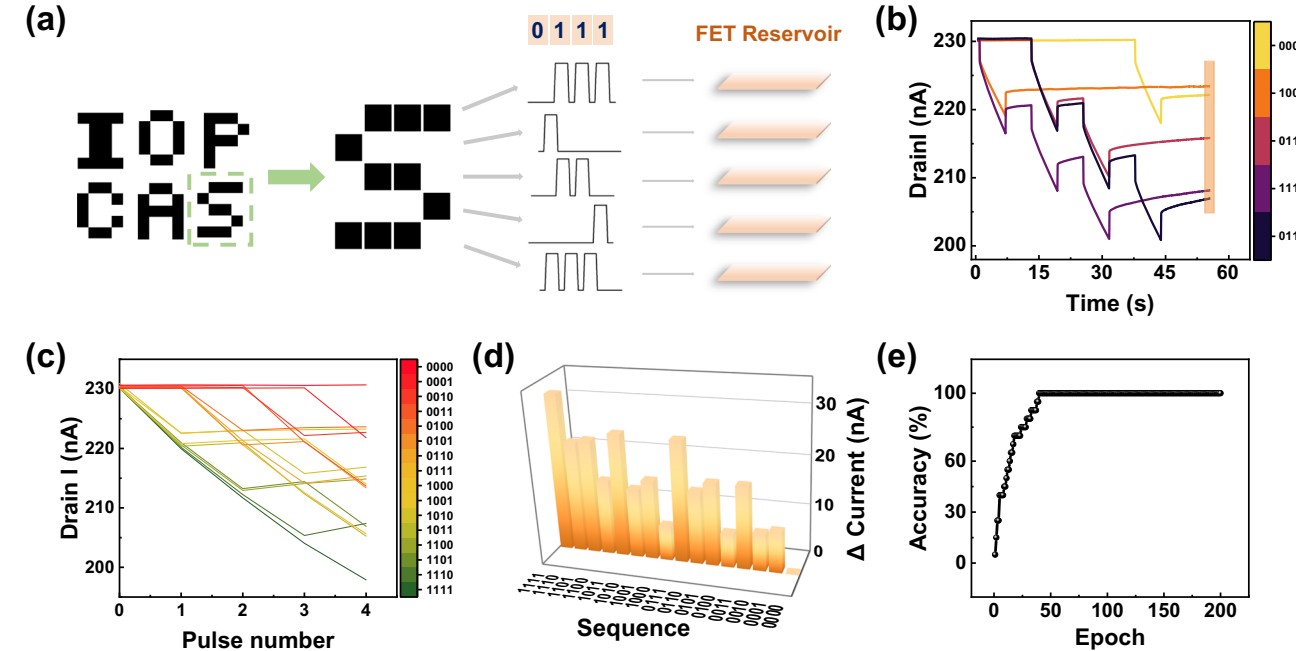

**Fig. 4 | Digital letter classification demonstration. a** Illustration showing the input process of the letters as pulse sequences into the reservoir. Each line of the letter pattern is divided into 4 pixels and then independently fed into the FET reservoir. The letter "S" is used as an example. **b** The evolution of drain current during the application of pulse sequences representing the letter "S". **c** All 16 kinds of sequence types and corresponding current outputs used for the letter classification. **d** The variation in drain current under different pulse sequences. **e** The evolution of classification precision during the training procedure, with 100% accuracy obtained in less than 100 epochs.

spoken recognition is to distinguish each independent word. As shown in Fig. 5a, b, we inferred a method to convert vocal signals into acoustic patterns which were later used as training and inference datasets[64]. First, the pristine sound wave was saved as a function of time and then transformed into a function frequency. For example, after recording the sound wave of the spoken word "lychee", we transform vocal information into a function of frequency via the fast Fourier transform (FFT), as recorded in the left panel of Fig. 5b.

The next step consisted of sampling the information by collecting 250 points from time and frequency signals. A total of 500 discrete signal data points were next transformed into an acoustic image (20×25), where each pixel contained a scale value within the range of 0–255. Five words, namely "lychee", "blueberry", "mango", "pomegranate", and "shaddock" spoken by four people were processed in our database. Finally, each data point was converted to binary data by setting a threshold value. For each acoustic pattern, 500 data points were recorded with either 1 or 0 values. When feeding the data into the reservoir, the inputs were divided into 125 groups, each composed of 4-timestream pulses, similar to our former application. Figure 5c demonstrates the results of every word spoken by four people, and the data from different people are separated by solid blue lines. In a real situation, we cannot guarantee absolute silence during the collection of information; therefore, noise would be inevitable. Thus, salt and pepper noise ($\delta = 0.05$) was added to simulate more authentic situations. The weight distribution before and after the training process in the readout layer is provided in Fig. 5d. After the training process, a more dispersive distribution of the weights was observed, and 100% accuracy was achieved in this acoustic pattern recognition task. The results further proved the good properties of our PRC system.

**Transistor-based reservoir system for dynamic classification and chaotic prediction**
The computational capability of the proposed PRC system was basically verified by the static and temporal classification tasks. To further investigate the potential of the reservoir system in processing time

series data, we performed two benchmark tasks to demonstrate the prediction of time series data by taking advantage of the aforementioned suitability. It is worth noting that for complex sequence information processing tasks, a large number of randomly interconnected nonlinear neuron nodes are required to build a reservoir capable of handling such tasks, which poses a significant challenge in terms of hardware implementation. Therefore, in order to overcome such difficulties, a mask technique has been employed to expand the input information and generate a large number of virtual nodes in the time domain[18]. A detailed description of the mask process can be found in the method section.

A task of waveform classification was used to test the temporal signal processing capability of the FET reservoir system[65]. As shown in Fig. 6a, the input sequence consisted of a random combination of sine and square waveforms. The target outputs were 0 and 1, representing the sine wave and square wave, respectively. After feeding the processed data into the reservoirs, different output current states representing the output of virtual nodes were sampled as the reservoir states. The time interval $\tau$ is defined as the total length of the voltage pulse sequence corresponding to each input data, i.e., the total duration of the sequence containing $M$ voltage pulses. During each time interval $\tau$, the output of the reservoir system consisted of a linear combination of all reservoir states. The schematic of the input signals and all reservoir states is shown in Supplementary Fig. 13. The amplitudes of the current states were distinguished after processing in the reservoir. Finally, the synaptic weights were trained through a simple linear regression algorithm and the classification results are summarized in Fig. 6b. Overall, the proposed reservoir system can correctly classify sine and square waves after training. In addition, the normalized root mean square error (NRMSE) was extremely low ($4.044 \times 10^{-8}$).

Chaotic system prediction can also be used as a benchmark to measure the performance of reservoir devices. The Hénon map was affirmed as a typical dynamic chaotic system and the task aimed to predict a new point $(x(n+1), y(n+1))$ based on the point $(x(n), y(n))$ in a

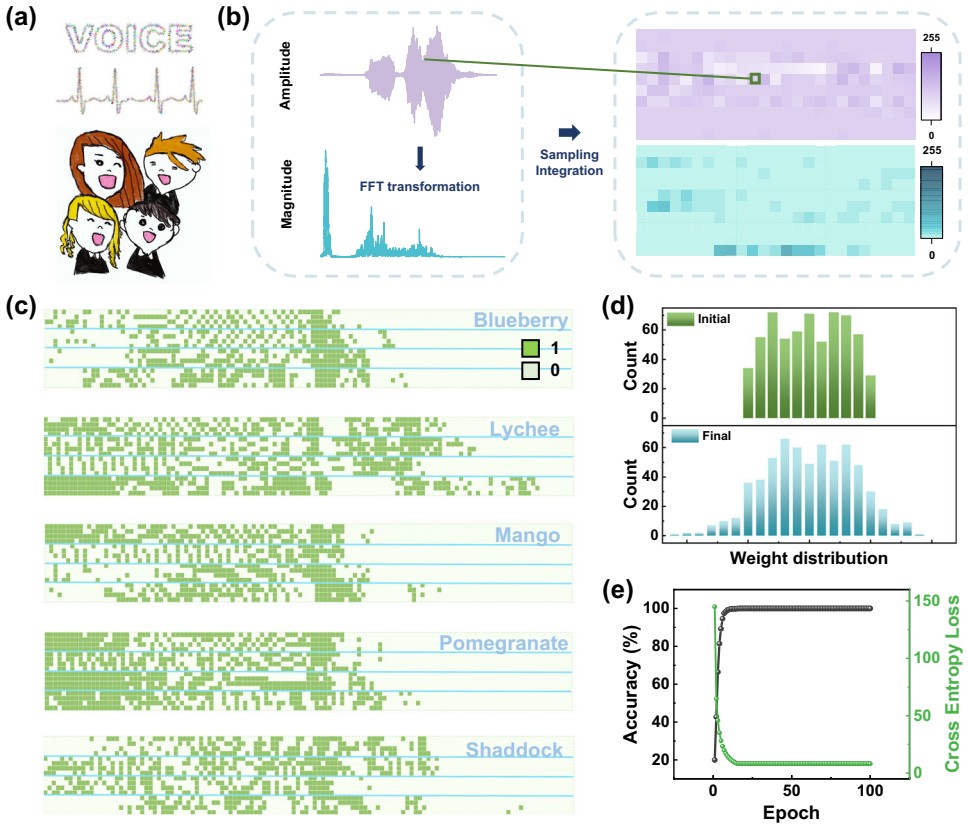

**Fig. 5 | Human voice recognition implemented by the FET-reservoir system.**
**a**, **b** Schematic representation of the voice input. The change in voice amplitude signal as a function of time converted to magnitude signal versus frequency through the FFT process. Two datasets extracted and integrated into a heat map range of 0–255. **c** All input data consisting of five English words spoken by four people. **d** The weight distributions before and after reservoir computing. **e** The classification accuracy of speech recognition with the proposed reservoir system along with the evolution of the cross-entropy loss.

nonlinear 2-D mapping on the plane[66,67]. The mapping can be defined as follows:

$$x(n+1) = y(n) - 1.4x(n)^2 \qquad (6)$$

$$y(n+1) = 0.3x(n) \qquad (7)$$

Through the combination of Eq. (6) and Eq. (7), the Hénon map could be deconstructed into a one-dimensional map, which could be described as $x(n+1) = 0.3x(n-1) - 1.4x(n)^2$. Therefore, a reservoir system capable of predicting $x(n+1)$ based on $x(n)$ was designed for the prediction of 2-D Hénon map. We created a dataset of 2000 data points, in which the first 1000 data points were used for training while the last 1000 data points were employed as inputs for testing. The input $x(n)$ sequence consisted of a linear mapping of the electrical pulses at amplitudes from 0 to 2 V. A mask process was also introduced as previously mentioned. During the input procedure, the input was multiplied by a mask with a length of 50. By applying different mask sequences, a reservoir system of $N$ parallel devices (here $N = 25$) can be simulated. The training process also adopted a linear regression algorithm. Figure 6c shows experimentally obtained outputs from the reservoir system after training and the solid line was taken as the target value obtained through theoretical computing. The results indicated that our reservoir system can solve the dynamic nonlinear problem with an NRMSE value of $5.85 \times 10^{-4}$. The prediction results demonstrated through a two-dimensional map in Fig. 6d revealed great consistency between the theoretical and simulation output based on the test data, proving the excellent performance of our device.

To explore the impact factors of prediction performance, we changed the maximum amplitude of applied voltage ($V_{max}$) and lengths of masks. As displayed in Fig. 6e, the prediction error varied as a function of both $V_{max}$ and mask length. The variation of the NRMSE as a function of the number of masks is shown in Supplementary Fig. 14. Therefore, the proper length and number of masks as well as the amplitude of the applied voltage were key parameters in obtaining better performances. An inappropriate mask length would induce weak feedback strength or insufficient states of the reservoir system, leading to poor prediction performance. In addition, we also investigated the effects of the pulse width and interval. Supplementary Fig. 15 shows the variation of the channel current with different pulse widths. The modulation range of the channel conductance becomes smaller as the pulse width becomes shorter. At this time, there are not enough oxygen vacancies in the LSMO films. To investigate the effect of the pulse width on the relaxation behavior, the channel current has been recorded at 100 s after the voltage pulse (Supplementary Fig. 15b). It can be seen that with a shorter pulse width, the variation of the channel current is closer to a linear process. In other words, the shorter the applied pulse widths, the worse the nonlinearity and volatility. For sequence tasks, the prediction error would increase with a shorter pulse width. We also performed a similar analysis on the pulse interval (Supplementary Fig. 15c, d). The extracted result shows that the modulation of the channel current becomes more linear with a shorter pulse interval. When the pulse interval is short, there is not enough time for the oxygen ions to move, so the relaxation phenomenon is not obvious.

For a more comprehensive analysis, we employed the nonlinear physical model mentioned in the manuscript (Eqs. (2)-(5)) based on the

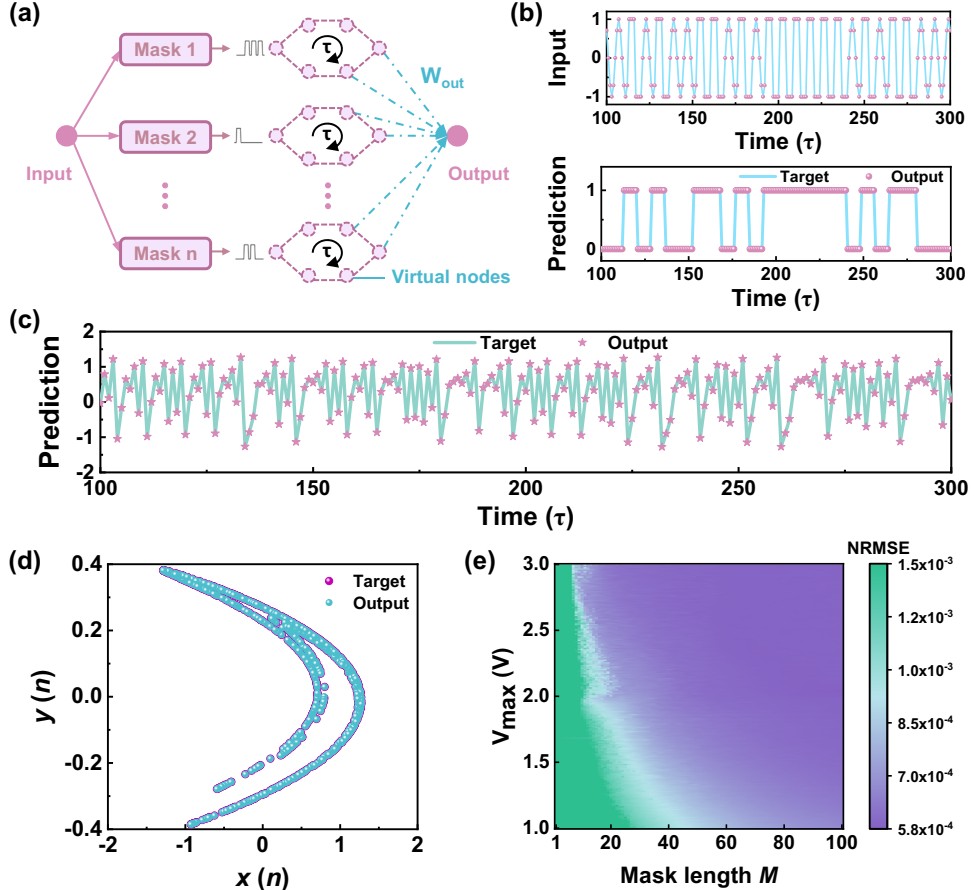

**Fig. 6 | Demonstration of time-dependent tasks, including wave classification and chaotic prediction. a** Schematic representation of a dynamic FET-based reservoir system. For a given input, the time multiplexing process is implemented by transforming the input to a temporal signal with a mask. **b** Random input consisting of sine and square waves. The lower panel shows the experimental classification output and target output by theoretical calculation. **c** The predicted results obtained by the FET devices. The pentagram mark represents the experimental output and the turquoise line represents the ideal target results. **d** Two-dimensional demonstration of the prediction output. **e** The accuracy of the prediction as a function of voltage range ($V_{max}$) and length of the mask ($M$). The color bar reveals the NRMSE.

device characteristics. Supplementary Fig. 16 illustrates the dependence of the Hénon map prediction error on variations in pulse width and pulse interval. It is evident that if the pulse width exceeds 10 s and the pulse interval exceeds 50 s, the errors consistently remain within a relatively small range. In this study, a relaxation time of 100 s was selected to comprehensively investigate the relaxation process after each pulse and to construct a nonlinear physical model. It should be noted that tasks with different time-scale may require different reservoir systems[68–70]. A physical system with long relaxation time like our device, may be appropriate for addressing long-time series tasks[68]. Furthermore, the intrinsic high performances of the proposed reservoir devices were also confirmed by conducting other chaotic prediction tasks. The predicted results of the Mackey-Glass oscillator and NARMA2 tasks are given in Supplementary Fig. 17. The NRMSE values were 0.008 and 0.096, respectively. Therefore, our PRC device showed extremely low prediction error compared to the former results (Supplementary Table 2) under all conditions. We also performed control experiments to prove the crucial role of reservoir computing when dealing with such time series tasks. We trained the input data directly with a linear model without reservoir processing, and the prediction performance of both tasks is much worse (Supplementary Figs. 18 and 19). The above results verified the feasibility of our proposed reservoir system in use as an effective hardware for reservoir computing.

## Discussion

In summary, a PRC system with a Mott-transistor gated by hafnium-based ferroelectric films was proposed. The unique coupled properties of polarization switching and ion migration in HZO thin films resulted in effective modulation of the conductance of the LSMO channels. We found intrinsic asymmetry dynamic process based on oxygen ions during the regulation of LSMO. The reversible transition between the PV and BM phases of LSMO induced nonlinear relaxation after the removal of the stimulation from the gate. The nonlinear and volatile properties of the device met the requirements for the high-performance reservoir. Taking advantage of the above properties, we have demonstrated a high-performance reservoir computing system. Pattern classification and speech recognition were achieved with 100% accuracy. Furthermore, the normalized root mean square error was extremely low for time-dependent tasks, such as wave classification and chaotic prediction.

The PRC system could potentially address the bottleneck problem of conventional computing systems as it operates as an in-memory framework. In this work, experimental proof-of-concept has been performed using a reservoir system based on oxygen ion dynamics. In the future, similar oxide reservoirs could be developed following the same principle as described in this study. The use of a three-terminal transistor structure allows for greater design flexibility and more effective reservoir tuning. Additionally, the design of specific

stimulation parameters should enable the achievement of reconfigurable functions that incorporate both volatile reservoirs and non-volatile synapses within the same device. The CMOS-compatible oxide-based PRC system paves the way for compact integration with standard computing platforms.

## Methods

### Sample preparation
The LSMO channel layer (~3.2 nm) and HZO (~10 nm) gate films were both deposited through pulsed laser deposition (PLD) method using a XeCl laser ($\lambda = 308$ nm) and the deposition temperatures of LSMO and HZO were 750 °C and 700 °C, respectively. The laser energy fluence was 1.75 J/cm$^2$, and the repetition rates used for both films were the same (2 Hz). The deposition rate of films was calibrated by X-ray reflection.

### Device fabrication
The channel layer was patterned through UV-lithography and Ar ion etching with dimensions of $10 \times 80$ μm. The source-drain Pt electrode was then deposited through magnetron sputtering with a thickness of 40 nm. Next, a 10 nm HZO film was prepared with a shadow mask. Finally, a 20 nm Pt gate electrode was patterned and deposited through UV-lithography and magnetron sputtering methods, respectively.

We also prepared electrodes with a smaller scale and the same heterostructure. Round Pt electrodes with a diameter of 30 μm and thickness of 50 nm were prepared through the same technology.

### Material characterization
X-ray diffraction (XRD) measurements were conducted using a high-resolution X-ray diffractometer from Rigaku Smartlab. $\theta$-$2\theta$ scanning was performed with a step width of 0.05° to characterize the structure and lattice parameters of the films. The wavelength of the X-ray is 1.5406 Å generated by the Cu anode tube.

The atomic force microscope (AFM) and piezoelectric force microscope (PFM) were both measured using the commercial Asylum Research MFP-3D probe microscope. Ir/Pt-coated conductive tips were used for PFM scanning. Dual A.C. resonance tracking (DART) mode were used to record phase and amplitude signals during the measurement. The drive amplitude was 1 V/1.5 V, and the switching voltage used for domain writing was ±9.5 V.

The polarization properties of Pt/HZO/LSMO/LSAT capacitors were measured by a precision multiferroic analyzer (RADIANT Tec. Inc.). Triangular pulses with a frequency of 10 kHz were applied.

### Electron microscopy
STEM imaging was conducted by a Cs-corrected JEOL JEM-ARM200CF NEOARM operated at 200 kV with a CEOS Cs corrector (CEOS GmbH, Heidelberg, Germany). HAADF-STEM images were recorded with collection semi-angles of 90-370 mrad. The EELS data were collected in dual-EELS mode to obtain both zero-loss spectra and core-loss spectra and recorded with a Gatan spectrometer, applying an energy dispersion of 0.1 eV per channel for Mn-$L$ and O-$K$ edges with a convergence semi-angle of 24 mrad. Core-loss EELS spectra were calibrated by corresponding zero-loss EELS before further analysis using Gatan Microscopy Suite Software.

### Electrical measurements
All electrical measurements of our FET devices were tested in a Lakeshore probe station with the Keithley 4200 semiconductor parameter analyzer. During electrical measurements, an amplitude of 0.1 V read voltage was applied to read the channel current. The measurements were conducted under ambient air at room temperature.

### Masking process
In the mask process, each data in the input sequence was multiplied by an $N \times M$ mask matrix, where $N$ is the number of masks (i.e., the number of reservoirs in parallel) and $M$ is the length of each mask[16]. In this task, $N$ and $M$ were set to 25 and 50 respectively, meaning that each input data was expanded into 25 data streams of length 50 and then fed into 25 parallel reservoirs in the form of voltage pulse sequences. The time interval $\tau$ is defined as the total length of the voltage pulse sequence corresponding to each input data, i.e., the total duration of the sequence containing $M$ voltage pulses. During each time interval $\tau$, the output of the reservoir system consisted of a linear combination of all reservoir states.

### Error analysis
The error of the output results was calculated by the normalized root mean squared error function, which can be defined as follows:

$$\text{NRMSE} = \sqrt{\frac{\sum_k \sum_i (p_i(k) - y_i(k))^2}{\sum_k \sum_i y_i^2(k)}}$$

where $p_i(k)$ is the experimentally predicted output and $y_i(k)$ is the target output.

## Data availability
Source data for the figures are provided as a Source data file. All relevant data within the Supplementary Information are available from the corresponding authors upon reasonable request. Source data are provided with this paper.

## Code availability
All code used in simulations supporting this article is available from the corresponding authors upon reasonable request.

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

## Acknowledgements
This work was supported by the National Key R&D Program of China (No. 2019YFA0308500 to K.J.), the National Natural Science Foundation of China (No. 12222414 to C.G., No. 12074416 to C.G., No. 11721404 to K.J., No. 12174437 to C.W.), the Youth Innovation Promotion Association of CAS (No. Y2022003 to C.G.).

## Author contributions
C.G. initiated the research. C.G. and K.J. supervised the project. The sample preparation, the device fabrication, and device measurements were performed by Z.L. with support from H.Z., M.Z., G.L. Q.Z., X.L. and L.G. contributed to STEM measurements. Simulations were performed by D.X. and M.Z. Z.L. and C.G. wrote the manuscript. M.H., D.S., C.W., K.J. and G.Y. participated in the discussion of the manuscript.

## Competing interests
The authors declare no competing interests.
