## [Peer Review File · Nature Communications]

REVIEWER COMMENTS

Reviewer #1 (Remarks to the Author):

The authors report a hafnia-based ferroelectric field effect transistor as reservoir for neuromorphic computing applications, demonstrating several of them. The transistor uses ferroelectric hafnia ($\text{Hf}_{0.5}\text{Zr}_{0.5}\text{O}_2$) as a gate oxide and a layer of $(\text{La,Sr})\text{MnO}_3$ as channel. Given that hafnia-based ferroelectric transistors have already been demonstrated as reservoirs to perform reservoir computing tasks (see ref. 27), the main novelty of the present work is to show a different channel material, where a different type of mechanism (in which the oxygen transport between the LSMO and the Hafnia layer determines the dynamics of the reservoir). I find the paper well written and the path to the demonstrations very clearly described, which it is appreciated and not often found in this type of multidisciplinary papers. However, in my opinion the work does not add sufficient novelty to the previous knowledge to deserve publication in a high impact journal as Nature Communications. This is true both from the concept point of view (as mentioned above) as from the materials viewpoint: the addition of the LSMO layer (compared with the Si channel used in ref, 27) is not new and the mechanisms that are involved because of this addition are assumed to be those already reported, in the references discussed in the paper (ref. 37 and 44). It is mentioned that “interaction between ferroelectric polarization and migration of oxygen was employed to manipulate the channel conductance.” But there are no new insights with respect to the nature of that interaction. Therefore, although I appreciate that this paper is useful for the field and should be published in some way, I doubt that this journal is the right place.

Reviewer #3 (Remarks to the Author):

The manuscript by Liu et al. reports a demonstration of gate-controlled oxygen ion dynamics for physical reservoir computing, by using a ferroelectric HZO gate and LSMO as the channel layer. Recently, three-terminal physical reservoir systems, mainly based on lithium ion and organic ions which are not compatible with CMOS processes, have attracted much attention. Oxygen ion injection and relaxation in oxides, demonstrated in this work, may be a promising way to fabricate novel three-terminal reservoir devices, because of its potential in CMOS compatibility. Combined with characterization results, the authors verify that the spontaneous migration of oxygen ions after electrical gating can lead to a non-linear dynamic procedure. Then, reservoir computing tasks were performed including static pattern recognition, voice recognition, waveform classification, and chaotic prediction. Overall, the manuscript is well-organized with solid experimental results and simulation output. This work is novel and interesting, and both the physical mechanism and the device prototype are demonstrated clearly. In principle, this work is appropriate for publication in

Nature Communications; however, the manuscript needs to be revised with minor corrections before being accepted. Below are my detailed suggestions and questions.

1. In the section of waveform classification and chaotic prediction, a time constant τ was introduced. Is this “ τ ” relative to the former mentioned constant in the double-exponential function(Eq.1)? If it is irrelevant, more detailed description of this constant should be added.
2. In Fig. 5c, according to the description in the manuscript, each word was spoken by four people, but it is not reflected in the figure. More detailed descriptions and clearer figures should be given.
3. The authors claimed that the prediction results in Fig. 6d revealed great consistency between the theoretical and experimental output. I suggest that the “experimental output” should be replaced by “simulation output based on tests data” to avoid misunderstanding.
4. The description of Fig. S6 is lacking in the manuscript. More detailed description of this Figure should be added.
5. In addition to the injection/diffusion of oxygen ions, the authors should focus on the role and influence of HZO ferroelectric polarization on channel conductance. More detailed discuss of this section should be given.
6. There are grammatical and writing errors, for example, the sentence “The reversibility phase transition between the PV and BM phases ...” in discussion section. Please check out the manuscript throughout and correct them.

Review of

Interface-type tunable oxygen ion dynamics for physical reservoir computing

Zhuohui Liu, Qinghua Zhang, Donggang Xie, Mingzhen Zhang, Xinyan Li, Hai Zhong, Ge Li, Meng He, Dashan Shang, Can Wang, Lin Gu, Guozhen Yang, Kuijuan Jin, and Chen Ge

Nature Communications, doi:-

Manuscript text

In this manuscript, the authors design and fabricate a dynamic transistor with a novel gate film. The principal advantage of doing so is that it imbues the transistor's current-voltage relationship with the two essential properties demanded of a reservoir - non-linearity of response and memory. Consequently they propose this device as a physical platform for computation and demonstrate its performance in several information processing tasks. Given the widespread current interest in reservoir computing, and the search for novel platforms for their implementation, I believe this paper is both timely and likely to be of some significance to researchers in the relevant disciplines. The results the authors obtain are an impressive proof-of-concept, demonstrating their fabricated device has the prerequisite physical properties to act as reservoir, and therefore the computational capacity to be applied to several standard problems which neural networks have been developed to address. For these reasons, I believe this work is likely to be suitable for publication, there are however several aspects of the manuscript that I believe should be improved before this, as detailed below.

First, I should state that my expertise does not lie in semi-conductor devices, so my comments and questions on the observed current dynamics and their origin may be elementary. I include them nevertheless on the grounds that proposing these HZO film gated transistors as a platform for reservoir computing is likely to attract interest from researchers who will similarly lack the background to answer these questions themselves. Firstly, the authors propose a mechanism wherein a positive voltage creates oxygen vacancies in the LSMO, and support this view with both STEM imaging of the heterostructure, and spectra of two of the edges in the LSMO. The effects of negative pulses on the film are also discussed, and based on the proposed model, it seems we would not observe the same spectral and STEM imaging features distinguishing pristine and pulsed LSMO. This raises the question of whether there exists an analogous measurement one could make to confirm the reduction of oxygen vacancies in the LSMO in this scenario, to further support the authors' explanation of the observed behaviours.

The results from the computational application of the transistor are certainly striking, but would benefit from a more detailed exposition of how they were obtained. For example, a "5×20 single layer ANN" (page 9 line 246) does not strike me as sufficient to describe the specific model being used and how it is trained. Another example would be the masking process that is referred to several times in the text. In particular, (page 12, line 310), the authors state that by applying these mask sequences "a reservoir system of N parallel devices (here N=25) can be simulated". I am not entirely clear on why this is the case. I think a more comprehensive explanation of how this feature - and the classification tasks more generally - are implemented would be a welcome addition to the methods. Ideally the authors would also make the analysis code they employed available on some repository, so that their methodology in obtaining the presented results can be properly scrutinised.

It should also be mentioned that the sections demonstrating the results of the various computational tasks employed appear under-referenced. As a non-exhaustive example, if one is going to talk about a Hénon map

there ought to be some citation of its use elsewhere in the literature. In the context of physical platforms for reservoir computing, the authors may also find (for example) the following papers relevant to their discussion: "Harnessing Disordered-Ensemble Quantum Dynamics for Machine Learning" Phys. Rev. Applied **8**, 024030 (2021), "Towards single-atom computing via high harmonic generation" EPJ Plus **138**: 123 (2023), "Photonic extreme learning machine by free-space optical propagation" Photonics Research **9**, 1446-1454 (2021). I would however consider these three suggestions as discretionary on the part of the authors.

A related issue that may be resolved with the inclusion of more detail concerns whether the obtained results are directly due to the reservoir properties of the transistor. For instance, that one could ask whether a linear model (classification or regression) trained directly on the data will have a similar accuracy to that where data is first processed by the reservoir. That is, it would strengthen the conclusions if it can be demonstrated that the performance of the classification tasks really is due to the reservoir properties of the transistor, rather than it simply acting as a novel encoding mechanism for a neural network.

The biggest concern I have is how long *physically* the collection of each reservoir state readout takes. That is, the physical relaxation dynamics place a bound on how quickly any given datum can be processed. As far as I can see, while the manuscript investigates how performance varies with gate voltage and mask length, pulse width and relaxation time before readout are not considered. Have the authors already experimented with this, and is the pulse width chosen as the smallest at which results can be reliably obtained? The manuscript ought to comment on this. Furthermore, while it would be unreasonable to demand this new hardware demonstrate tangible superiority over standard platforms at present, it would be helpful for the the authors to outline how they envision the eventual implementation of this transistor such that this specialised hardware will be able to outperform conventional computing in relevant tasks.

Finally, a few smaller points:

- while HZO is defined in the abstract, this is lacking in the main body of the text. A definition should be added at its first appearance in text (page 4, line 91)
- Fig S5. shows the change in dynamics as pulse width and pulse number are varied. What is perhaps unclear is the pulse width when varying the number of pulses. Judging by eye it appears to be using a 5s width, but this is likely worth noting.
- Function (x) is a slightly confusing way to refer to the equations in text. Ordinarily one would reference these with the abbreviation Eq. (x)
- Eqs.(3) and Eq.(5) depend on two different (presumably) time variables t_{modu} and t_{relax} . I cannot find a definition for these in the text, and how they are distinguished.
- Speaking of equations, is it necessary to present Eqs. (6) and (7) as two separate equations? Am I missing anything if I were to suggest one could describe the mapping recursively as

$$x(n+1) = 0.3x(n-1) - 1.4x^2(n) \quad (1)$$

- Lastly, while the manuscript is largely coherent and comprehensible, there exists some slightly awkward and unnatural phraseology within the text. It may therefore benefit from having a native speaker provide some copy-editing assistance.

Manuscript ID: NCOMMS-23-26950

Title: Interface-type tunable oxygen ion dynamics for physical reservoir computing

We thank all the reviewers for valuable comments regarding our research paper. Each of your insights have served to strengthen our manuscript. We have carefully revised the manuscript according to your constructive suggestions. Provided below is our detailed response to each comment raised.

Reviewer #1 (Remarks to the Author):

The authors report a hafnia-based ferroelectric field effect transistor as reservoir for neuromorphic computing applications, demonstrating several of them. The transistor uses ferroelectric hafnia ($\text{Hf}_{0.5}\text{Zr}_{0.5}\text{O}_2$) as a gate oxide and a layer of $(\text{La,Sr})\text{MnO}_3$ as channel. Given that hafnia-based ferroelectric transistors have already been demonstrated as reservoirs to perform reservoir computing tasks (see ref. 27), the main novelty of the present work is to show a different channel material, where a different type of mechanism (in which the oxygen transport between the LSMO and the Hafnia layer determines the dynamics of the reservoir). I find the paper well written and the path to the demonstrations very clearly described, which it is appreciated and not often found in this type of multidisciplinary papers. However, in my opinion the work does not add sufficient novelty to the previous knowledge to deserve publication in a high impact journal as Nature Communications. This is true both from the concept point of view (as mentioned above) as from the materials viewpoint: the addition of the LSMO layer (compared with the Si channel used in ref, 27) is not new and the mechanisms that are involved because of this addition are assumed to be those already reported, in the references discussed in the paper (ref. 37 and 44). It is mentioned that “interaction between ferroelectric polarization and migration of oxygen was employed to manipulate the channel conductance.” But there are no new insights with respect to the nature of that interaction. Therefore, although I appreciate that this paper is useful for the field and should be published in some way, I doubt that this journal is the right place.

Response:

Thank you very much for your hard-work in reviewing our manuscript. We appreciate the reviewer for the judgements “*I find the paper well written and the path to the demonstrations very clearly described, which it is appreciated and not often found in this type of multidisciplinary papers.*”. We agree with the reviewer that the interaction between ferroelectric polarization and migration of oxygen ions is not a newly discovered phenomenon, however, we did not intend to claim novelty on the discovery of the polarization-charge coupled effect and ferroelectric reservoirs.

In fact, the novelty we would like to highlight here is the first application of interface-type oxygen ion dynamics in the physical reservoir computing. This represents an important advance of significance to neuromorphic computing community, as

elaborated below.

(1) First, we would like to elucidate the growing significance of physical reservoir system within the state-of-the-art neuromorphic community. Reservoir computing, distinguished by its reduced training expenditure compared to conventional feedforward networks, serves as an effective framework for processing temporal data^{R1-R3}. Notably, reservoir computing offers a viable solution for predicting chaotic systems, a type of task that is challenging for conventional computing methodologies. Reservoir computing systems implemented through software rely on conventional CMOS technology, which suffers from high power consumption and integration problems. More recently, studies have focused on the hardware implementation of reservoir systems^{R4-R6}. As a computational framework in physics, physical reservoir computing (PRC) which contains inherent dynamics may overcome the bottleneck dilemma of former methods^{R7}. This revolutionizes the progress towards next-generation neuro-inspired computers, and the use of previously unexplored materials for computation. However, designing and implementing a suitable dynamic system is by no means an easy feat. Currently, it is a matter of urgency and importance to identify a physical reservoir suitable for the requirements of a particular task. As the Reviewer #2 commented, *“Given the widespread current interest in reservoir computing, and the search for novel platforms for their implementation, I believe this paper is both timely and likely to be of some significance to researchers in the relevant disciplines.”*

(2) Our study presents a novel PRC system utilizing gate-tunable oxygen ion dynamics, which has not been proposed before. We validated that this dynamic process satisfies the criteria for high-performance reservoir computing^{R8}. It is worthy to note that the work mentioned by the reviewer is based on a pure ferroelectric transistor^{R9}. In that study, ferroelectric polarization dynamics in an FeFET lead to the nonlinearity of the flowing current. In contrast, we proposed a completely different ion-dynamics mechanism. The HZO film in our study plays a more important role as an oxygen-ion conductor than as a ferroelectric gate. Our work provides a prototype demonstration of a PRC system based on oxygen ion dynamics. In the future, other analogous oxide systems could be developed based on the same principle in this work. Furthermore, due to the different dynamic mechanism, the relaxation times of these two systems are fundamentally different. It should be noted that different sets of tasks require distinct memory properties of the reservoir^{R10}. Some tasks may require a long memory while some may require a short memory^{R10-R12}. Compared to other transistors based on lithium^{R13} and organic ions^{R14}, oxygen-ion-based reservoirs exhibit full compatibility with the complementary metal-oxide-semiconductor (CMOS) technology, and are environmentally friendly^{R15}. As the Reviewer #3 commented, *“Oxygen ion injection and relaxation in oxides, demonstrated in this work, may be a promising way to fabricate novel three-terminal reservoir devices, because of its potential in CMOS compatibility.”*

(3) Moreover, the proposed device exhibits excellent performance. We have

implemented the processing of several benchmark tasks, which include static pattern classification, speech recognition, and chaotic prediction. The recognition accuracy of letters and spoken words remains at 100% even after introducing noise. In the prediction tasks of different chaotic systems such as Hénon map^{R16}, Mackey-Glass oscillator^{R17} and NARMA2 system^{R18}, our reservoir system exhibits ultra-low prediction error. The above-mentioned results highlight the great potential of our device as an efficient reservoir system.

Therefore, our study not only demonstrates the first type of oxygen-ion-based dynamic reservoir transistor with excellent performance, but also opens up a new solution for the development of reliable and efficient hardware for reservoir computing system with CMOS compatibility. We believe that the novelty and significance of our study are sufficient to warrant its publication in Nature Communications. We are looking forward to a refreshed view of this reviewer.

Corresponding references

- R1. Triefenbach, F. *et al.* J.-P. Phoneme recognition with large hierarchical reservoirs. *Adv. Neural Inf. Process. Syst.* **23**, 2307–2315 (2010).
- R2. Jaeger, H. & Haas, H. Harnessing nonlinearity: predicting chaotic systems and saving energy in wireless communication. *Science* **304**, 78–80 (2004).
- R3. Maass, W. *et al.* Real-time computing without stable states: a new framework for neural computation based on perturbations. *Neural Comput.* **14**, 2531–2560 (2002)
- R4. Moon J. *et al.* Temporal data classification and forecasting using a memristor-based reservoir computing system. *Nature Electronics.* **2**, 480-487 (2019).
- R5. Milano G. *et al.* In materia reservoir computing with a fully memristive architecture based on self-organizing nanowire networks. *Nature Materials* **21**, 195-202 (2022).
- R6. Zhong Y. *et al.* Dynamic memristor-based reservoir computing for high-efficiency temporal signal processing. *Nat. Commun.* **12**, 408 (2021).
- R7. Hirose A, *et al.* Physical reservoir computing: Possibility to resolve the inconsistency between neuro-AI principles and its hardware. *Aust J Intell Inf Process Syst* **16**, 49-55 (2019).
- R8. Tanaka G. *et al.* Recent advances in physical reservoir computing: A review. *Neural Networks* **115**, 100-123 (2019).
- R9. Toprasertpong K. *et al.* Reservoir computing on a silicon platform with a ferroelectric field-effect transistor. *Communications Engineering* **1**, 21 (2022).
- R10. Cucchi M, *et al.* Hands-on reservoir computing: a tutorial for practical implementation. *Neuromorphic Computing and Engineering* **2**, 032002 (2022).
- R11. Schuecker J, *et al.* Optimal Sequence Memory in Driven Random Networks. *Physical Review X* **8**, 041029 (2018).
- R12. Durstewitz D. *et al.* Neurocomputational models of working memory. *Nature Neuroscience* **3**, 1184-1191 (2000).
- R13. Nishioka D. *et al.* Edge-of-chaos learning achieved by ion-electron-coupled dynamics in an ion-gating reservoir. *Sci. Adv.* **8**, eade1156 (2022).
- R14. Cucchi M. *et al.* Reservoir computing with biocompatible organic electrochemical networks for brain-inspired biosignal classification. *Sci. Adv.* **7**, eabh0693 (2021).

- R15. Huang M. *et al.* Electrochemical Ionic Synapses: Progress and Perspectives. *Adv. Mater.* 2205169, (2023).
- R16. Hénon, M. The Theory of Chaotic Attractors. 94–102 (2004).
- R17. Jaeger H *et al.* Harnessing Nonlinearity: Predicting Chaotic Systems and Saving Energy in Wireless Communication. *Science* **304**, 78-80 (2004).
- R18. Atiya AF. *et al.* New results on recurrent network training: unifying the algorithms and accelerating convergence. *IEEE transactions on neural networks* **11**, 697-709 (2000).

Changes made:

To describe the background of reservoir computing more clearly, we changed “Reservoir computing can efficiently be used to solve time-dependent tasks than conventional feedforward networking owing to various advantages, such as easy training, low hardware overhead, and implementation in electrical and optical devices. The high-efficient reservoir systems would require nonlinear and dynamic responses to distinguish time-series input data.” to “Reservoir computing can efficiently be used to solve time-dependent tasks than conventional feedforward networking owing to various advantages, such as easy training and low hardware overhead. Physical reservoirs which contain intrinsic nonlinear dynamic processes could serve as next-generation dynamic computing systems.” [Abstract]

To describe the background of physical reservoir computing more clearly, we added “Initially, software-level reservoir computing based on existing CMOS platforms was used to perform relevant dynamic tasks¹⁴. However, CMOS devices do not have intrinsic characteristics of dynamic response, so the processing of nonlinear dynamic tasks requires the combination with complex algorithms and large-scale integrated devices, which leads to unnecessary hardware consumption¹⁵. Fortunately, this problem can be resolved through the implementation of the physical reservoir computing (PRC) with spontaneous nonlinear processes¹⁵. Due to the highly adaptive and flexible dynamic characteristics in the physical systems, more reliable PRC hardware with smaller size is expected to further enhance the performance of dynamic processing¹⁶.” before “In summary, the implementation of reservoir computing requires some particular features.” [paragraph 1, page 3]

We added “The PRC system could potentially address the bottleneck problem of conventional computing systems as it operates as an in-memory framework.” [paragraph 1, page 15]

Corresponding references were added in the revised manuscript.

14. Jang YH, Kim W, Kim J, *et al.* Time-varying data processing with nonvolatile memristor-based temporal kernel. *Nature Communications* **12**, 5727 (2021).
15. Qi Z, Mi L, Qian H, *et al.* Physical Reservoir Computing Based on Nanoscale Materials and Devices. *Advanced Functional Materials* 2306149, (2023).
16. Zhang Z, Zhao X, Zhang X, *et al.* In-sensor reservoir computing system for latent fingerprint recognition with deep ultraviolet photo-synapses and memristor array. *Nature*

Communications **13**, 6590 (2022).

To further illustrate that our work is based on oxygen ion dynamics, we added several sentences in the revised manuscript:

“The HZO film plays a more important role in this work as an oxygen-ion conductor than as a ferroelectric gate.” [paragraph 1, page 9]

“In this work, experimental proof-of-concept has been performed using a reservoir system based on oxygen ion dynamics. In the future, similar oxide reservoirs could be developed following the same principle as described in this study.” [paragraph 1, page 15]

Reviewer #2 (Remarks to the Author):

In this manuscript, the authors design and fabricate a dynamic transistor with a novel gate film. The principal advantage of doing so is that it imbues the transistor's current-voltage relationship with the two essential properties demanded of a reservoir - non-linearity of response and memory. Consequently, they propose this device as a physical platform for computation and demonstrate its performance in several information processing tasks. Given the widespread current interest in reservoir computing, and the search for novel platforms for their implementation, I believe this paper is both timely and likely to be of some significance to researchers in the relevant disciplines. The results the authors obtain are an impressive proof-of-concept, demonstrating their fabricated device has the prerequisite physical properties to act as reservoir, and therefore the computational capacity to be applied to several standard problems which neural networks have been developed to address. For these reasons, I believe this work is likely to be suitable for publication, there are however several aspects of the manuscript that I believe should be improved before this, as detailed below.

Response:

We greatly appreciate the reviewer for the positive recommendation and constructive comments. The point-by-point responses to the reviewer's comments are listed in the following.

Point 1.

First, I should state that my expertise does not lie in semi-conductor devices, so my comments and questions on the observed current dynamics and their origin may be elementary. I include them nevertheless on the grounds that proposing these HZO film gated transistors as a platform for reservoir computing is likely to attract interest from researchers who will similarly lack the background to answer these questions themselves. Firstly, the authors propose a mechanism wherein a positive voltage creates oxygen vacancies in the LSMO, and support this view with both STEM imaging of the heterostructure, and spectra of two of the edges in the LSMO. The effects of negative pulses on the film are also discussed, and based on the proposed model, it seems we would not observe the same spectral and STEM imaging features distinguishing pristine and pulsed LSMO. This raises the question of whether there exists an analogous measurement one could make to confirm the reduction of oxygen vacancies in the LSMO in this scenario, to further support the authors' explanation of the observed behaviours.

Response:

Thanks for the reviewer's valuable suggestion. Following your suggestion, we conducted the related STEM experiments to confirm the reduction of oxygen vacancies in the negatively pulsed LSMO film.

As stated in the manuscript (Paragraph 1, Page 8), the LSMO film prefers an oxygen-

rich state, resulting in fewer oxygen vacancies in the pristine film^{R19}. We agree with the reviewer that we may not observe a clear change in the STEM and EELS results between the pristine and negatively pulsed films. Therefore, we first performed a positive voltage pulse on our heterostructure device (LSMO/HZO/Pt), and then applied a negative voltage pulse. Here, we define this sample as “Negatively Pulsed”. The obtained STEM images and EELS results are presented and analyzed in the following section.

From the EELS spectra, it can be observed that the *L*-edge of Mn in the negatively pulsed LSMO film shifts towards the high-energy direction compared to the positively pulsed film (Fig. R1). Additionally, the pre-peak of the *K*-edge of O reappears. These spectral results verify the reduction of oxygen vacancies^{R20, R21}. The atomic resolution image of negatively pulsed LSMO is shown in Fig. R2. The lattice parameters were also calculated, and the decreased value of the out-of-plane lattice parameters further suggests that the lattice expansion caused by oxygen vacancy has been restored. (Fig. R3)

Fig. R1. STEM-EELS spectra of Mn *L*-edge and O *K*-edge in the pristine and pulsed LSMO films.

Fig. R2. STEM image of the negatively pulsed heterostructure.

Fig. R3. Lattice analysis based on the STEM measurements. (a) Variation of out-of-plane lattice parameter in the pristine and pulsed LSMO films. (b) The in-plane spacing along [010] direction.

Corresponding references

R19. Hu K. *et al.* Atomic-scale observation of strain-dependent reversible topotactic transition in $\text{La}_{0.7}\text{Sr}_{0.3}\text{MnO}_x$ films under an ultra-high vacuum environment. *Materials Today Physics* **29**, (2022).

R20. Varela M. *et al.* Atomic-resolution imaging of oxidation states in manganites. *Physical Review B* **79**, 085117 (2009).

R21. Li Z. *et al.* Interface and Surface Cation Stoichiometry Modified by Oxygen Vacancies in Epitaxial Manganite Films. *Adv. Funct. Mater.* **22**, 4312-4321 (2012).

Changes made:

1. We added the atomic-resolution STEM image and EELS spectra of negatively pulsed LSMO films. Correspondingly, we added the definition of negatively pulsed sample. The updated Fig. 2 is shown below.

We have added description about the negatively pulsed sample.

“The amplitude of voltage pulse is ± 3 V. Here, the defined “Negatively Pulsed” sample is firstly pulsed by a positive voltage and then pulsed by the negative voltage.” [paragraph 2, page 7]

We changed “*Ex-situ* observations of atomic lattice in the heterostructures before and after the application of 3 V positive pulses are depicted in Fig. 2a and 2b, respectively.” to “*Ex-situ* observations of atomic lattice in the heterostructures before and after the application of pulses are depicted in Fig. 2a-2c, respectively.” [paragraph 2, page 7]

We changed “The amplified images of enclosed regions with a square in Fig. 2a and 2b are provided in Fig. 2c and 2d, respectively.” to “The amplified images of enclosed regions with a square in Fig. 2a-2c are provided in Fig. 2d-2f, respectively.” [paragraph 2, page 7]

Fig. 2 Structural characteristics of the LSMO channel before and after pulse stimulation. **a-c** STEM image of the **a)** pristine, **b)** positively pulsed and **c)** negatively pulsed heterostructure. **d-f** Enlarged views of the selected areas of the boxes in **a)**, **b)** and **c)**. **g, h** STEM-EELS spectra of **g)** Mn- L_2 and Mn- L_3 edges and **h)** O-K edges for the pristine and pulsed LSMO layers.

2. We have updated Supplementary Fig. 6 with Fig. R3 and added description about the experimental result.

We added “The lattice parameters of negatively pulsed sample were also calculated, and the decreased value of the out-of-plane lattice parameters suggests that the lattice expansion caused by oxygen vacancy has been restored.” [paragraph 2, page 7]

Supplementary Figure 6. Lattice analysis based on the STEM measurements. **a** Variation of out-of-plane lattice parameter in the pristine and pulsed LSMO films. **b** The in-plane spacing along [010] direction.

3. In order to make the relevant descriptions logically, we changed the order of Supplementary Fig. 6 and Supplementary Fig. 7. We updated the renumbered Supplementary Fig. 7, and added the corresponding description in the revised manuscript and Supplementary Information.

We added “Supplementary Fig. 7 shows the depth analysis of STEM-EELS spectra in pristine and pulsed LSMO films. The same spectral characteristics at different depths reveal that the modulation by oxygen ions can be effective throughout the film.” [paragraph 1, page 8]

Supplementary Figure 7. Depth analysis of STEM-EELS. Variation of the Mn L -edge from top to the bottom interfaces of pristine and pulsed LSMO films **a** pristine, **c** positively pulsed, **e** negatively pulsed. Variation of the O K -edge from top to the bottom interfaces of pristine and pulsed LSMO films. **b** pristine, **d** positively pulsed, **f** negatively pulsed.

4. We added two sentences and a new Supplementary Fig. 8.

“Supplementary Figure 8a and 8b summarizes the peak positions of Mn L_3 edge and area ratio of the L_3/L_2 edge peaks for the pristine and pulsed films. The shift of the peak towards higher energy after negative voltage stimulation confirms the oxidation of Mn^{3+} .” [paragraph 1, page 8]

Supplementary Figure 8. Peak analysis of Mn L_3 edge before and after the stimulation. **a** A significant shift towards lower energy after positive voltage stimulation indicates the reduction of Mn^{4+} . The shift of the peak towards higher energy after negative voltage stimulation confirms the oxidation of Mn^{3+} . **b** The ratio of peak area L_3/L_2 in the pristine and pulsed LSMO films.

Point 2.

The results from the computational application of the transistor are certainly striking, but would benefit from a more detailed exposition of how they were obtained. For example, a “5×20 single layer ANN” (page 9 line 246) does not strike me as sufficient to describe the specific model being used and how it is trained. Another example would be the masking process that is referred to several times in the text. In particular, (page 12, line 310), the authors state that by applying these mask sequences “a reservoir system of N parallel devices (here N=25) can be simulated”. I am not entirely clear on why this is the case. I think a more comprehensive explanation of how this feature - and the classification tasks more generally - are implemented would be a welcome addition to the methods. Ideally the authors would also make the analysis code they employed available on some repository, so that their methodology in obtaining the presented results can be properly scrutinised.

Response:

We thank the reviewer for this valuable suggestion. Following your suggestion, we will elaborate in more detail about two examples.

1. In the tasks of pattern recognition, the letters with original size of 20 pixels were divided into 5 rows, each of them contains 4 pixels. The sequence representing each row of pixels is inputted into the device as a series of voltage pulses. The outputs of channel current are shown in Fig. 4. As a result, a letter pattern was compressed after processed by the reservoirs, so the number of nodes in the input layer of the artificial neural network is 5. The number of nodes in the output layer, on the other hand, is equal to the classes of categories of the final classification results (20). We then used a single-layer fully connected neural network with a size of 5×20 to perform the letter

recognition task. The Softmax function was selected as the activation function of the output layer, and the weights were updated based on the backpropagation algorithm^{R22-R24}.

2. For complex sequence information processing tasks, a large number of randomly interconnected nonlinear neuron nodes are required to build a reservoir capable of handling such tasks, which is undoubtedly a challenge in terms of hardware implementation. Therefore, to overcome such difficulty, a mask process was used to expand the input information and generate a large number of virtual nodes in the time domain^{R25-R27}. Each data in the input sequence was multiplied by an $N \times M$ mask matrix, where N is the number of masks (i.e., the number of reservoirs in parallel) and M is the length of each mask. In this task, N and M were set to 25 and 50 respectively, meaning that each input data was expanded into 25 data streams with length of 50 and then fed into 25 parallel reservoirs in the form of voltage pulse sequences. Various output current states representing the output of virtual nodes were then sampled as the reservoir states.

Corresponding references

R22. Yihao Yang *et al.* Spin-Filtering Ferroelectric Tunnel Junctions as Multiferroic Synapses for Neuromorphic Computing. *ACS Applied Materials & Interfaces* **12** (50), 56300-56309, (2020).

R23. Li, G. *et al.* Flexible VO₂ Films for In-Sensor Computing with Ultraviolet Light. *Adv. Funct. Mater.* **32**, 2203074 (2022).

R24. Ahmed, T. *et al.* Fully Light-Controlled Memory and Neuromorphic Computation in Layered Black Phosphorus. *Adv. Mater.* **33**, 2004207 (2021).

R25. Zhong Y. *et al.* Dynamic memristor-based reservoir computing for high-efficiency temporal signal processing. *Nat. Commun.* **12**, 408 (2021).

R26. Chen, Z. *et al.* All-ferroelectric implementation of reservoir computing. *Nat. Commun.* **14**, 3585 (2023).

R27. Hu, W. *et al.* Distinguishing artificial spin ice states using magnetoresistance effect for neuromorphic computing. *Nat. Commun.* **14**, 2562 (2023).

3. Following your suggestions, we have uploaded our codes to the “Github”.

The links is: <https://github.com/IOP-L03/Physical-Reservoir-Computing>

Changes made:

We changed the description of training method during the classification tasks with added details in the revised manuscript:

We changed “During the classification, the output current signals of the final reservoir states were fed into a 5×20 single layer ANN for readout. The read-out output consisted of input voltages multiplied by the conductance weights.” to “During the classification, a single-layer fully connected neural network with a size of 5×20 was used to perform the letter recognition task. The Softmax function was selected as the activation function of the network, and the weights were updated based on the backpropagation algorithm.”

[paragraph 2, page 10]

We changed “The static and temporal classification tasks basically verified the computing capacity of the proposed physical reservoir system. To further explore the potential of reservoir system in time series data processing. Hence, we performed two benchmark tasks to demonstrated the prediction of temporal data by taking advantage of the said suitability.” to “The computational capability of the proposed PRC system was basically verified by the static and temporal classification tasks. To further investigate the potential of the reservoir system in processing time series data, we performed two benchmark tasks to demonstrate the prediction of time series data by taking advantage of the aforementioned suitability. It is worth noting that for complex sequence information processing tasks, a large number of randomly interconnected nonlinear neuron nodes are required to build a reservoir capable of handling such tasks, which poses a significant challenge in terms of hardware implementation. Therefore, in order to overcome such difficulties, a mask technique has been employed to expand the input information and generate a large number of virtual nodes in the time domain¹⁸. A detailed description of the mask process can be found in the method section.”

[paragraph 1, page 12]

We changed “Each data in the input sequence was multiplied by a mask, and each input data was extended to a 50-pulse timeframe. Various output current states were then recorded as virtual nodes. During each time interval τ , the output of the reservoir system consisted of a linear combination of all reservoir states. The schematic of input signals and all reservoir states are shown in Supplementary Fig. 11. The amplitude of current states was distinguished after processing in the reservoir.” to “After feeding the processed data into the reservoirs, different output current states representing the output of virtual nodes were sampled as the reservoir states. The time interval τ is defined as the total length of the voltage pulse sequence corresponding to each input data, i.e., the total duration of the sequence containing M voltage pulses. During each time interval τ , the output of the reservoir system consisted of a linear combination of all reservoir states. The schematic of the input signals and all reservoir states is shown in Supplementary Fig. 10. The amplitudes of the current states were distinguished after processing in the reservoir.” [paragraph 2, page 12]

Corresponding references were added in the revised manuscript.

18. Zhong Y, Tang J, Li X, *et al.* Dynamic memristor-based reservoir computing for high-efficiency temporal signal processing. *Nat. Commun.* **12**, 408 (2021).

We added the description of masking process to the section of “Method”:

Masking process

In the mask process, each data in the input sequence was multiplied by an $N \times M$ mask matrix, where N is the number of masks (i.e., the number of reservoirs in parallel) and M is the length of each mask¹⁶. In this task, N and M were set to 25 and 50 respectively,

meaning that each input data was expanded into 25 data streams of length 50 and then fed into 25 parallel reservoirs in the form of voltage pulse sequences. The time interval τ is defined as the total length of the voltage pulse sequence corresponding to each input data, i.e., the total duration of the sequence containing M voltage pulses. During each time interval τ , the output of the reservoir system consisted of a linear combination of all reservoir states. [Method, page 17]

Point 3.

It should also be mentioned that the sections demonstrating the results of the various computational tasks employed appear under-referenced. As a non-exhaustive example, if one is going to talk about a Hénon map there ought to be some citation of its use elsewhere in the literature. In the context of physical platforms for reservoir computing, the authors may also find (for example) the following papers relevant to their discussion: “Harnessing Disordered-Ensemble Quantum Dynamics for Machine Learning” *Phys. Rev. Applied* **8**, 024030 (2021), “Towards single-atom computing via high harmonic generation” *EPJ Plus* **138**: 123 (2023), “Photonic extreme learning machine by free-space optical propagation” *Photonics Research* **9**, 1446-1454 (2021). I would however consider these three suggestions as discretionary on the part of the authors.

Response:

We are very grateful to the reviewer for the valuable suggestions on our manuscript. We appreciate the reviewer to recommend three literatures on physical platforms for reservoir computing, which have been added as new references in the revised manuscript. Besides, we have added several relevant references in the sections of related computational tasks to improve the manuscript.

Changes made:

1. We changed a sentence and added related references:

“In the last decades, many studies have so far demonstrated the implementation of reservoir systems with applications in speech recognition¹⁹, chaotic prediction^{20, 21} electric consumption prediction²², fingerprint identification¹⁶ based on two-terminal memristors^{19, 23, 24}, spintronic oscillators²⁵, programmable logic gate arrays²⁶, photonic module devices²⁶⁻²⁸, and quantum devices²⁹”. [paragraph 1, page 4]

Corresponding references

27. Pierangeli D, Marcucci G, Conti C. Photonic extreme learning machine by free-space optical propagation. *Photon Res* **9**, 1446-1454 (2021).

28. McCaul G, Jacobs K, Bondar DI. Towards single atom computing via high harmonic generation. *The European Physical Journal Plus* **138**, 123 (2023).

29. Fujii K, Nakajima K. Harnessing Disordered-Ensemble Quantum Dynamics for Machine Learning. *Physical Review Applied* **8**, 024030 (2017).

2. We added a related reference:

“A waveform classification task was used to test the temporal signal processing capability of the FET reservoir system⁶⁵.” [paragraph 1, page 12]

Corresponding references

65. Paquot Y, Duport F, Smerieri A, *et al.* Optoelectronic Reservoir Computing. *Scientific Reports* **2**, 287 (2012).

3. We added related references:

“The Hénon map was affirmed as a typical dynamic chaotic system and the task aimed to predict a new point in a nonlinear 2-D mapping on the plane^{66,67}.” [paragraph 3, page 12]

Corresponding references

66. Hénon M. *The Theory of Chaotic Attractors*. Springer, New York (2004).

67. Rodan A, Tino P. Minimum Complexity Echo State Network. *IEEE transactions on neural networks* **22**, 131-144 (2011).

4. We added related references in the revised Supplementary Information:

“The NRMSE values of predicting Mackey-Glass oscillator¹ and NARMA² tasks were 0.008 and 0.096, respectively.”

Corresponding references

1. Jaeger H, Haas H. Harnessing Nonlinearity: Predicting Chaotic Systems and Saving Energy in Wireless Communication. *Science* **304**, 78-80 (2004).

2. Atiya AF, Parlos AG. New results on recurrent network training: unifying the algorithms and accelerating convergence. *IEEE transactions on neural networks* **11**, 697-709 (2000).

Point 4.

A related issue that may be resolved with the inclusion of more detail concerns whether the obtained results are directly due to the reservoir properties of the transistor. For instance, that one could ask whether a linear model (classification or regression) trained directly on the data will have a similar accuracy to that where data is first processed by the reservoir. That is, it would strengthen the conclusions if it can be demonstrated that the performance of the classification tasks really is due to the reservoir properties of the transistor, rather than it simply acting as a novel encoding mechanism for a neural network.

Response:

Thanks for the reviewer’s valuable comments on our manuscript. Following your suggestions, we trained directly on the data using a linear model when performing relevant tasks. Detailed results are as follows.

1. For the classification of letters, the accuracy obtained is around 80 % if we use a linear model. To further explain this result, the letters “A” and “H” have been taken as examples (Fig. R4). For these two letters, the number of “1” and “0” inputs are identical

in the same line. Since the output of the linear model is only related to the number of input pulses, there is no difference between the letters “A” and “H” after linear reservoir processing. Therefore, three pairs of letters cannot be correctly classified. (“A” and “H”; “K”, “O” and “Y”; “B” and “R”)

Fig. R4. Classification accuracy obtained using a linear model.

2. For time series data processing, the input data is multiplied by a mask matrix and then fed into the linear reservoirs. Subsequently, different output current states representing the output of virtual nodes were sampled as the reservoir states. Since the output of the linear model loses order information, the NRMSE increases significantly compared to that of the nonlinear model. For waveform classification and chaotic system prediction, the NRMSE increases significantly from 1.99×10^{-8} to 0.932 and from 5.85×10^{-4} to 1.06, respectively. (Fig. R5 and Fig. R6). The results suggest that the nonlinear models outperform linear models when processing time series information, which is a fact widely corroborated^{R28-R31}.

Fig. R5. Comparison of waveform prediction results between reservoir nonlinear model and linear model. (a) input waveform, (b) prediction result with PRC pre-processing, (c) prediction result with linear model.

Fig. R6. Comparison of Hénon map prediction results between reservoir nonlinear model (a), (c) and linear model (b), (d).

Corresponding references

R28. Kunwar P. *et al.* Linear and nonlinear modeling approaches for urban air quality prediction, *Science of The Total Environment*, **426**, 244-255, (2012).

R29. Nikita Basant. *et al.* Linear and nonlinear modeling for simultaneous prediction of dissolved oxygen and biochemical oxygen demand of the surface water — A case study, *Chemometrics and Intelligent Laboratory Systems*, **104**(2), (2010).

R30. Pedersen, D. R. *et al.* Direct Comparison of Muscle Force Predictions Using Linear and Nonlinear Programming. *ASME. J Biomech Eng.* **109**(3): 192–199. (1987).

R31. Roman M. *et al.* Comparison of linear and nonlinear calibration models based on near infrared (NIR) spectroscopy data for gasoline properties prediction, *Chemometrics and Intelligent Laboratory Systems*, **88**(2): 183-188, (2007).

Changes made:

We added description about classification result with a linear model: “In order to verify the performance of the classification tasks is indeed due to the reservoir properties of the transistor, we performed the same tasks using a linear model without reservoir computing processing. The classification accuracy decreased to 80 % (Supplementary Fig. 12).” [paragraph 2, page10]

We added a new Supplementary Fig. 12:

Supplementary Figure 12. Classification accuracy obtained through a linear model. Letters with the same number of “1” or “0” inputs in the same row cannot be distinguished by the linear model without processing by reservoir computing.

We added Fig. R5 and Fig. R6 as updated Supplementary Fig. 18 and Supplementary Fig. 19 in the Supplementary Information.

We added description about waveform and chaotic prediction results with a linear model: “We also performed control experiments to prove the crucial role of reservoir computing when dealing with such time series tasks. We trained the input data directly with a linear model without reservoir processing, and the prediction performance of

both tasks is much worse (Supplementary Fig. 18 and Supplementary Fig. 19).”
 [paragraph 2, page 14]

Supplementary Figure 18. Comparison of waveform prediction results between reservoir nonlinear model and linear model. a input waveform, **b** prediction result with PRC pre-processing, **c** prediction result with linear model.

Supplementary Figure 19. Comparison of Hénon map prediction results between

reservoir nonlinear model and linear models. a, c results obtained through nonlinear model, **b, d** results obtained through linear model.

Point 5.

The biggest concern I have is how long physically the collection of each reservoir state readout takes. That is, the physical relaxation dynamics place a bound on how quickly any given datum can be processed. As far as I can see, while the manuscript investigates how performance varies with gate voltage and mask length, pulse width and relaxation time before readout are not considered. Have the authors already experimented with this, and is the pulse width chosen as the smallest at which results can be reliably obtained? The manuscript ought to comment on this.

Furthermore, while it would be unreasonable to demand this new hardware demonstrate tangible superiority over standard platforms at present, it would be helpful for the authors to outline how they envision the eventual implementation of this transistor such that this specialised hardware will be able to outperform conventional computing in relevant tasks.

Response:

We are very grateful to the reviewer for his/her constructive suggestion on our manuscript.

1. Following your kind suggestion, we tested the device with different pulse widths and intervals. As shown in Fig. R7a, the range of channel conductance becomes smaller as the pulse width becomes shorter. In this case, there are not enough oxygen vacancies in the LSMO films. To investigate the effect of the pulse width on the relaxation behavior, the channel current has been recorded at 100 seconds after the voltage pulse (Fig. R7b). It can be seen that with a shorter pulse width, the variation of the channel current is closer to a linear process. In other words, the shorter the applied pulse width, the worse the nonlinearity and volatility. For sequence tasks, the prediction error would increase with a shorter pulse width. We also performed a similar analysis on the pulse interval, as shown in Fig. R7c and Fig. R7d. The extracted result shows that the modulation of the channel current becomes more linear with a shorter pulse interval. When the pulse interval is short, there is not enough time for the oxygen ions to move, so the relaxation process is not obvious.

For a more comprehensive analysis, we employed the nonlinear physical model mentioned in the manuscript (Eq. (2)-Eq. (5), page 9) based on the device characteristics to investigate the dependence of the Hénon map prediction error on variations in pulse width and pulse interval (Fig. R8). It is evident that if the pulse width exceeds 10 seconds and the pulse interval exceeds 50 seconds, the errors consistently remain within a relatively small range. Therefore, a relaxation time of 100 s was selected in this study to comprehensively investigate the relaxation process after each pulse and to construct a nonlinear physical model.

The system presented in this work is primarily intended to serve as a prototype demonstration of the reservoir computing system based on a novel principle. For the hardware realization of reservoir computing, how to select a suitable nonlinear system for a given task needs to be further studied in depth, as there are many nonlinearities available in nature^{R32}. Specifically, tasks with different time-scale may require different reservoir systems^{R10-R12, R33}. For example, phoneme classification requires only a short time-scale memory of less than 1 s, whereas language translation would require a longer memory of multiple seconds^{R13}. In consequence, a physical system with long relaxation time like our device, may be appropriate for addressing long-time series tasks.

Fig. R7. Variation of the channel current under various positive pulse. **a** different pulse widths. **c** different intervals between each pulse. The amplitude of voltage is 2.5 V. **b**, **d** is extracted drain current data from **a** and **c**, respectively.

Fig. R8. NRMSE of Hénon map prediction as a function of pulse width and pulse interval.

2. We sincerely appreciate the reviewer for his/her comments on the outlook of this field. Here, we try our best to analyze and compare different approaches considering the time series tasks.

The conventional artificial neural network (ANN) based on feedforward neural network fails to extract sequential data from the input data. Therefore, a combination of ANN and other methods such as Autoregressive Integrated Moving Average (ARIMA) is necessary to solve dynamic tasks^{R34}. As a contrast, the recurrent neural network (RNN) comprises feedback loops among hidden neurons. Therefore, the RNN outputs rely on the current inputs and the previous states of the neurons. This allows the RNN to identify temporal correlations in the data^{R35}. The conventional RNN utilizes the backpropagation through time (BPTT) algorithm, but its training process is computationally expensive and slow. Researchers have discovered that only the output connection weight exhibits significant changes during RNN training via the BPTT algorithm. Inspired by this, echo state network (ESN)^{R36}, liquid state machine (LSM)^{R37} were proposed, later being referred as reservoir computing^{R38}. Based on this, the training process of RNN has been greatly simplified since only the weights connecting the reservoir and the output need to be trained.

Currently, some software-based reservoirs have been used for temporal sequence prediction^{R39}, phoneme recognition^{R40} and motion control^{R41}. Software computing operates on established CMOS platforms. However, CMOS devices lack intrinsic dynamic response characteristics. Therefore, the processing of dynamic tasks requires a combination of complex algorithms and large integrated devices, resulting in excess hardware consumption.

In nature, many physical systems contain spontaneous dynamic processes^{R32} which are suitable for the reservoir computing system. Therefore, combining spontaneous nonlinear processes in physical systems with relevant algorithms is expected to optimize the performance of dynamic task^{R42}. Besides, the reservoir based on physical principles are more stable and reliable than conventional calculation methods^{R43}. It is worthy to mention that due to the highly adaptive and flexible characteristics of many materials, physical reservoir computing (PRC) can adapt to more sorts of tasks and scenarios^{R44}. In recent years, the exploration of reservoir hardware has received widespread attention.

Next, we discuss the potential advantages of the proposed device compared to other PRC devices. Three-terminal transistors provide more freedoms for device designing due to the additional input terminals. The transistor-based PRC devices can be tuned to different computational tasks by applying various voltage pulses. This can meet the potential needs of realizing multi-functional PRC system. The material combination we proposed can also be used to construct ferroelectric field-effect transistors. By tuning the external simulation conditions, it is expected to realize non-volatile memory in the same device^{R45}. It is thus feasible to achieve a compact computing system through the incorporation of a reconfigurable device with both volatile reservoir and non-volatile synaptic functions. In terms of integration, the oxide materials are of CMOS-compatible potential and environmentally friendly^{R46}.

In conclusion, our work provides a new approach to construct a suitable dynamic system for reservoir computing. In contrast to conventional computing techniques, PRC systems exhibit repeatability, adjustability and expansibility. By embedding CMOS-compatible reservoir hardware on microchips, the size and power requirements of computational system can be drastically reduced.

Corresponding references

R10. Cucchi M, *et al.* Hands-on reservoir computing: a tutorial for practical implementation. *Neuromorphic Computing and Engineering* **2**, 032002 (2022).

R11. Schuecker J, *et al.* Optimal Sequence Memory in Driven Random Networks. *Physical Review X* **8**, 041029 (2018).

R12. Durstewitz D. *et al.* Neurocomputational models of working memory. *Nature Neuroscience* **3**, 1184-1191 (2000).

R32. Tanaka G, *et al.* Recent Advances in Physical Reservoir Computing: A Review. *Neural Networks*. **115**, 100-123 (2018).

R33. Liu K. *et al.* An optoelectronic synapse based on α -In₂Se₃ with controllable temporal dynamics for multimode and multiscale reservoir computing. *Nature Electronics* **5**, 761-773 (2022).

R34. Wang L *et al.* An ARIMA-ANN Hybrid Model for Time Series Forecasting. *Syst. Res.* **30**, 244-259 (2013).

R35. Lukoševičius M. *et al.* Reservoir computing approaches to recurrent neural network training. *Computer Science Review* **3**, 127-149 (2009).

- R36. Jaeger H. The "echo-state" approach to analysing and training recurrent neural networks. (2001).
- R37. Maass W. *et al.* Real-Time Computing Without Stable States: A New Framework for Neural Computation Based on Perturbations. *Neural Computation* **14**, 2531-2560 (2002).
- R38. Verstraeten D. *et al.* An experimental unification of reservoir computing methods. *Neural Networks* **20**(3), 391-403 (2007).
- R39. Jaeger H, *et al.* Harnessing Nonlinearity: Predicting Chaotic Systems and Saving Energy in Wireless Communication. *science* 304, 78-80 (2004).
- R40. Triefenbach F. *et al.* Phoneme recognition with large hierarchical reservoirs. In: Proceedings of the 23rd International Conference on Neural Information Processing Systems. *Curran Associates Inc.* **2**, (2010).
- R41. Joshi P. *et al.* Movement Generation and Control with Generic Neural Microcircuits. In: Biologically Inspired Approaches to Advanced Information Technology. *Springer Berlin Heidelberg* (2004).
- R42. Jang YH *et al.* Time-varying data processing with nonvolatile memristor-based temporal kernel. *Nature Communications* **12**, 5727 (2021).
- R43. Qi Z. *et al.* Physical Reservoir Computing Based on Nanoscale Materials and Devices. *Advanced Functional Materials*, 2306149, (2023).
- R44. Zhang Z. *et al.* In-sensor reservoir computing system for latent fingerprint recognition with deep ultraviolet photo-synapses and memristor array. *Nature Communications* **13**, 6590 (2022).
- R45. Halter M. *et al.* Back-End, CMOS-Compatible Ferroelectric Field-Effect Transistor for Synaptic Weights. *ACS Applied Materials & Interfaces* **12**, 17725-17732 (2020).
- R46. Hong X. *et al.* Emerging ferroelectric transistors with nanoscale channel materials: the possibilities, the limitations. *Journal of Physics: Condensed Matter* **28**, 103003 (2016).

Changes made:

1. We added Fig. R7 and Fig. R8 as updated Supplementary Fig. 15 and Supplementary Fig. 16, respectively.

We added description about supplementary experiments:

“In addition, we also investigated the effects of the pulse width and interval. Supplementary Fig. 15 shows the variation of the channel current with different pulse widths. The modulation range of the channel conductance becomes smaller as the pulse width becomes shorter. At this time, there are not enough oxygen vacancies in the LSMO films. To investigate the effect of the pulse width on the relaxation behavior, the channel current at 100 seconds after the voltage pulse (Supplementary Fig. 15b). It can be seen that with a shorter pulse width, the variation of the channel current is closer to a linear process. In other words, the shorter the pulse widths applied, the worse the nonlinearity and volatility. As a result, the prediction error increases with a shorter pulse width. We also performed a similar analysis on the pulse interval (Supplementary Fig. 15c-d). The extracted result shows that the modulation of the channel current becomes more linear with a shorter pulse interval. When the pulse interval is short, there is not enough time for the oxygen ions to move, so that the relaxation phenomenon is not obvious.”

[paragraph 2, page 13]

“For a more comprehensive analysis, we employed the nonlinear physical model mentioned in the manuscript (Eq. (2)-Eq. (5)) based on the device characteristics. Supplementary Fig. 16 illustrates the dependence of the Hénon map prediction error on variations in pulse width and pulse interval. It is evident that if the pulse width exceeds 10 seconds and the pulse interval exceeds 50 seconds, the errors consistently remain within a relatively small range. In this study, a relaxation time of 100 s was selected to comprehensively investigate the relaxation process after each pulse and to construct a nonlinear physical model. It should be noted that tasks with different time-scale may require different reservoir systems⁶⁸⁻⁷⁰. A physical system with long relaxation time like our device, may be appropriate for addressing long-time series tasks⁶⁸.” [paragraph 1, page 14]

Corresponding references

68. Cucchi M, *et al.* Hands-on reservoir computing: a tutorial for practical implementation. *Neuromorphic Computing and Engineering* **2**, 032002 (2022).
69. Schuecker J, *et al.* Optimal Sequence Memory in Driven Random Networks. *Physical Review X* **8**, 041029 (2018).
70. Durstewitz D. *et al.* Neurocomputational models of working memory. *Nature Neuroscience* **3**, 1184-1191 (2000).

Supplementary Figure 15. Variation of the channel current under various positive pulse. a different pulse widths. **c** different intervals between each pulse. The amplitude of voltage is 2.5 V. **b, d** is extracted drain current data from **a** and **c**, respectively.

Supplementary Figure 16. NRMSE of Hénon map prediction as a function of pulse width and pulse interval.

2. To describe the background of reservoir computing more clearly, we changed “Reservoir computing can efficiently be used to solve time-dependent tasks than conventional feedforward networking owing to various advantages, such as easy training, low hardware overhead, and implementation in electrical and optical devices. The high-efficient reservoir systems would require nonlinear and dynamic responses to distinguish time-series input data.” to “Reservoir computing can efficiently be used to solve time-dependent tasks than conventional feedforward networking owing to various advantages, such as easy training and low hardware overhead. Physical reservoir which contains intrinsic nonlinear dynamic processes could serve as next-generation dynamic computing systems.” [Abstract]

To further outlook the development of our device, we added several sentences in the discussion of revised manuscript:

“The PRC system could potentially address the bottleneck problem of conventional computing systems as it operates as an in-memory framework. In this work, experimental proof-of-concept has been performed using a reservoir system based on oxygen ion dynamics. In the future, similar oxide reservoirs could be developed following the same principle as described in this study.” [paragraph 1, page 15]

“Additionally, the design of specific stimulation parameters should enable the achievement of reconfigurable functions that incorporate both volatile reservoirs and nonvolatile synapses within the same device. The CMOS-compatible oxide-based PRC system paves the way for compact integration with standard computing platforms.” [paragraph 1, page 15]

Point 6.

Finally, a few smaller points:

• **Point 6.1**

while HZO is defined in the abstract, this is lacking in the main body of the text. A definition should be added at its first appearance in text (page 4, line 91)

Response:

Thanks for the reviewer's valuable comments on our manuscript. We added the definition in the revised manuscript.

Changes made:

We changed "HZO" to " $\text{Hf}_{0.5}\text{Zr}_{0.5}\text{O}_2$ (HZO)" in paragraph 3, page 4.

• **Point 6.2**

Fig S5. shows the change in dynamics as pulse width and pulse number are varied. What is perhaps unclear is the pulse width when varying the number of pulses. Judging by eye it appears to be using a 5s width, but this is likely worth noting.

Response:

Thanks for the reviewer's valuable comments on our manuscript. We apologize for the unspecific description. The width of pulses is indeed 5 s, we have added statement in the revised Supplementary Information.

Changes made:

We added "The width of each pulse is 5 s." in the revised Supplementary Information.

• **Point 6.3**

Function (x) is a slightly confusing way to refer to the equations in text. Ordinarily one would reference these with the abbreviation Eq. (x)

Response:

Thanks for the reviewer's valuable comments on our manuscript. Following your kind suggestions, we have made relevant modifications in the revised manuscript.

Changes made:

We changed "Function (2)" to "Eq. (2)"; "Function (3)" to "Eq. (3)"; "Function (4)" to "Eq. (4)"; "Function (5)" to "Eq. (5)" and "Function (2)-(5)" to "Eqs. (2)-(5)" [paragraph 2, page 9]

We changed "Function (1) and Function (2)" to "Eq. (6) and Eq. (7)" [paragraph 3, page 12]

• **Point 6.4**

Eqs. (3) and Eq. (5) depend on two different (presumably) time variables t_{modu} and t_{relax} . I cannot find a definition for these in the text, and how they are distinguished.

Response:

Thanks for the reviewer's valuable comments. We are sorry for missing this detail. In the revised manuscript, we added the definition about these variables.

Changes made:

We added "Among them, variable t_{modu} is the duration of applying voltage, i.e., modulation time, and t_{relax} is the relaxation time after removing the gate voltage." after "The coefficient k , b , and c are all constants." [paragraph 2, page 9]

• Point 6.5

Speaking of equations, is it necessary to present Eqs. (6) and (7) as two separate equations? Am I missing anything if I were to suggest one could describe the mapping recursively as $x(n+1) = 0.3x(n-1) - 1.4x(n)^2$

Response:

Thanks for your valuable comments. Indeed, the Hénon map could be deconstructed into a one-dimensional map, defined as $x(n+1) = 0.3x(n-1) - 1.4x(n)^2$. However, it is worth noting that the Hénon map is actually a two-dimensional chaotic system. If the combined formula is used to describe this system, it may result in an inaccurate statement and fail to declare the meaning of $y(n)$ in Fig. 6d^{R16, R47}. Predicting $x(n+1)$ based solely on $x(n)$ is a simplification of predicting the chaotic system and does not imply a lack of correlation between $x(n+1)$ and $y(n)$. For a proper depiction of the 2-D chaotic system, it may be clearer to define it using two distinct equations. We have included more detailed descriptions in our revised manuscript.

Corresponding references

R16. Hénon, M. The Theory of Chaotic Attractors. 94–102 (2004).

R47. Rodan A. *et al.* Minimum Complexity Echo State Network. *IEEE transactions on neural networks* **22**, 131-144 (2011).

Changes made:

We changed "The Hénon map was affirmed as a typical dynamic chaotic system and the task aimed to predict a new point in a nonlinear 2-D mapping on the plane" to "The Hénon map was affirmed as a typical dynamic chaotic system and the task aimed to predict a new point $(x(n+1), y(n+1))$ based on the point $(x(n), y(n))$ in a nonlinear 2-D mapping on the plane^{66,67}." [paragraph 3, page 12]

We changed "Through iteration of functions (1) and (2), a formula related to only the variation X can be obtained. Therefore, the task was simplified to predict the $f(x(n+1))$ based on $f(x(n))$. We implemented a dataset of 2000 data points, in which the first 1000 data points were used for training while the last 1000 data points were employed as inputs for testing." to "Through the combination of Eq. (6) and Eq. (7), the Hénon map could be deconstructed into a one-dimensional map, which could be described as $x(n+1) = 0.3x(n-1) - 1.4x(n)^2$. Therefore, a reservoir system capable of predicting

$x(n+1)$ based on $x(n)$ was designed for the prediction of 2-D Hénon map. We created a dataset of 2000 data points, in which the first 1000 data points were used for training while the last 1000 data points were employed as inputs for testing.” [paragraph 1, page 13]

Corresponding references

66. Hénon, M. The Theory of Chaotic Attractors, 94–102 (2004).

67. Rodan A. *et al.* Minimum Complexity Echo State Network. *IEEE transactions on neural networks* **22**, 131-144 (2011).

• **Point 6.6**

Lastly, while the manuscript is largely coherent and comprehensible, there exists some slightly awkward and unnatural phraseology within the text. It may therefore benefit from having a native speaker provide some copy-editing assistance.

Response:

Thanks for your valuable suggestions. We apologize for the improper English expression in our manuscript. We have involved editing service from “Research Square” (a partner of Nature Springer) to assist with language corrections. We have revised the text to adhere to academic writing principles, including logical structure, precise word choice, and grammatical correctness. We really hope that the flow and language level have been substantially improved.

Reviewer #3 (Remarks to the Author):

The manuscript by Liu et al. reports a demonstration of gate-controlled oxygen ion dynamics for physical reservoir computing, by using a ferroelectric HZO gate and LSMO as the channel layer. Recently, three-terminal physical reservoir systems, mainly based on lithium ion and organic ions which are not compatible with CMOS processes, have attracted much attention. Oxygen ion injection and relaxation in oxides, demonstrated in this work, may be a promising way to fabricate novel three-terminal reservoir devices, because of its potential in CMOS compatibility. Combined with characterization results, the authors verify that the spontaneous migration of oxygen ions after electrical gating can lead to a non-linear dynamic procedure. Then, reservoir computing tasks were performed including static pattern recognition, voice recognition, waveform classification, and chaotic prediction. Overall, the manuscript is well-organized with solid experimental results and simulation output. This work is novel and interesting, and both the physical mechanism and the device prototype are demonstrated clearly. In principle, this work is appropriate for publication in Nature Communications; however, the manuscript needs to be revised with minor corrections before being accepted. Below are my detailed suggestions and questions.

Response:

We greatly appreciate the reviewer for the positive recommendation and valuable comments. The detailed responses to the reviewer are given as follows.

Point 1.

In the section of waveform classification and chaotic prediction, a time constant τ was introduced. Is this “ τ ” relative to the former mentioned constant in the double-exponential function(Eq.1)? If it is irrelevant, more detailed description of this constant should be added.

Response:

Thanks for the reviewer’s valuable comments. The constant “ τ ” mentioned in two place is irrelevant. In the double-exponential function (Eq. (1)), τ is time constant used for describing the dynamic relaxation process. Meanwhile, constant “ τ ” defined in the section of waveform classification and chaotic prediction represents the total length of the voltage pulse sequence corresponding to each input data. To avoid confusion, we have added a sentence in the revised manuscript to clarify this point.

Changes made:

We added a sentence:

“The time interval τ is defined as the total length of the voltage pulse sequence corresponding to each input data, that is, the total duration of the sequence containing M voltage pulses.” [paragraph 2, page 12]

Point 2.

In Fig. 5c, according to the description in the manuscript, each word was spoken by four people, but it is not reflected in the figure. More detailed descriptions and clearer figures should be given.

Response:

Thanks for the reviewer’s valuable comments. Following your kind suggestion, we added dividing lines to separate the data from different individuals, and the following are related changes in the revised manuscript.

Changes made:

We updated Fig. 5 and added “Fig. 5c demonstrated the results of every word spoken by four people, and the data from different people are separated by solid blue lines.” [paragraph 2, page 11]

Updated Fig. 5

Point 3.

The authors claimed that the prediction results in Fig. 6d revealed great consistency between the theoretical and experimental output. I suggest that the “experimental output” should be replaced by “simulation output based on tests data” to avoid misunderstanding.

Response:

Thanks for the reviewer's valuable suggestion. We revised our description in the manuscript based on your comments.

Changes made:

We changed "The prediction results demonstrated through a two-dimension map in Fig. 6d revealed great consistency between the theoretical and experimental output, proving an excellent performance of our device" to "The prediction results demonstrated through a two-dimension map in Fig. 6d revealed great consistency between the theoretical and simulation output based on the tests data, proving an excellent performance of our device." [paragraph 1, page 13]

Point 4.

The description of Fig. S6 is lacking in the manuscript. More detailed description of this Figure should be added.

Response:

We appreciate the reviewer's valuable comments and apologize for the missing description. In the revised manuscript, we added a few sentences about Supplementary Fig. 6 (revised Supplementary Fig. 7).

Changes made:

To make the relevant description logical, we have changed the order of Supplementary Fig. 7 to Supplementary Fig. 6:

"Supplementary Figure. 7 shows the depth analysis of STEM-EELE spectra in pristine and pulsed LSMO films. The same spectral characteristics at different depths reveal that the modulation by oxygen ions can be effective throughout the film." [paragraph 1, page 8]

Point 5.

In addition to the injection/diffusion of oxygen ions, the authors should focus on the role and influence of HZO ferroelectric polarization on channel conductance. More detailed discuss of this section should be given.

Response:

We appreciate the reviewer for this valuable comment. Next, we discuss the contributions of the ion migration and ferroelectric polarization to the channel conductance.

On one hand, the modulation of the channel conductance through ferroelectric polarization is typically non-volatile in the FeFET devices^{R48}, because of the spontaneous polarization property. However, our device exhibits significant volatile response under positive stimulation, suggesting that the change of channel conductance is primarily caused by oxygen ion migration. Additionally, we conducted transfer curve measurements with a faster rate. Figure R9 depicts the results of the voltage scanning.

The pulse width was set to 18 ms and the voltage range was -4.9 V to 5.2 V. In this scenario, the modulation of the channel current was mainly attributed to the ferroelectric polarization, as the ions did not have sufficient time to migrate due to the short pulse width. It can be seen that the modulation range of the channel current is much smaller than that of the slower measurement. Therefore, ion migration plays a more important role than ferroelectric polarization in our wide pulse experiment.

On the other hand, the regulation of Mott-material is very difficult through ferroelectric polarization^{R49}. We listed several works based on the ferroelectric field effect transistors (Table R1). The modulation range of the channel conductance was compared. To define the modulation range uniformly, following equation has been used (Eq. (R1)), where ΔI donates the difference between I_{max} and I_{min} at zero gate voltage. The limited modulation range of Mott material is apparent even when gated by a ferroelectric material with a significant polarization value. In contrast, the migration of oxygen ions can efficiently modulate the LSMO channel in our work.

$$\text{Modulation range} = \frac{\Delta I}{I_{min}} \quad (R1)$$

In summary, the HZO film plays a more important role in this work as an oxygen-ion conductor

Fig. R9. Transfer curve of our FET device with a faster scanning rate. The pulse width was 18 ms.

Channel material	Ferroelectric gate	Polarization value	Modulation range
GdBa ₂ Cu ₃ O _{7-x} (2 nm) ^{R50}	Pb(Zr,Ti)O ₃ (PZT)	10 μC/cm ²	15 %
LSMO (4 nm) ^{R51}	Pb(Zr,Ti)O ₃ (PZT)	48 μC/cm ²	50 %
LSMO (4 nm) ^{R52}	Pb(Zr,Ti)O ₃ (PZT)	80 μC/cm ²	50 %
LaNiO ₃ (~1.2 nm) ^{R53}	Pb(Zr,Ti)O ₃ (PZT)	30 μC/cm ²	150 %
LSMO (3.2 nm) ^{This work}	Hf _{0.5} Zr _{0.5} O ₂ (HZO)	20 μC/cm ²	450 %

Table R1. Comparison of modulations with ferroelectric gating.

Corresponding references

R48. Mathews S. *et al.* Ferroelectric Field Effect Transistor Based on Epitaxial Perovskite Heterostructures. *Science* **276**, 238-240 (1997).

R49. Hong X. *et al.* Emerging ferroelectric transistors with nanoscale channel materials: the possibilities, the limitations. *Journal of Physics: Condensed Matter* **28**, 103003 (2016).

R50. Ahn CH. *et al.* Electrostatic Modulation of Superconductivity in Ultrathin GdBa₂Cu₃O_{7-x} Films. *Science* **284**, 1152-1155 (1999).

R51. Hong X. *et al.* Examining the screening limit of field effect devices via the metal-insulator transition. *Applied Physics Letters* **86**, 142501 (2005).

R52. Vaz CAF. *et al.* Origin of the Magnetoelectric Coupling Effect in Pb(Zr_{0.2}Ti_{0.8})O₃/La_{0.8}Sr_{0.2}MnO₃ Multiferroic Heterostructures. *Physical Review Letters* **104**, (2010).

R53. Marshall MSJ. *et al.* Conduction at a Ferroelectric Interface. *Physical Review Applied* **2**, 051001 (2014).

Changes made:

We added a few sentences and added a related reference:

“It should be noted that the modulation of the channel conductance through ferroelectric polarization is typically non-volatile due to its spontaneous polarization property^{53, 63}. However, our device shows an obvious volatile response under electrical gating, which indicates that the observed response characteristics are dominated by the migration of oxygen ions. In other words, the HZO film plays a more important role in this work as an oxygen-ion conductor than as a ferroelectric gate.” [paragraph 1, page 9]

Corresponding references

53. Mathews S, Ramesh R, Venkatesan T, *et al.* Ferroelectric Field Effect Transistor Based on Epitaxial Perovskite Heterostructures. *Science* **276**, 238-240 (1997).

63. Guo R, You L, Lin W, *et al.* Continuously controllable photoconductance in freestanding BiFeO₃ by the macroscopic flexoelectric effect. *Nature Communications* **11**, 2571 (2020).

Point 6.

There are grammatical and writing errors, for example, the sentence “The reversibility

phase transition between the PV and BM phases ...” in discussion section. Please check out the manuscript throughout and correct them.

Response:

Thanks for the reviewer’s valuable comments. We sincerely apologize for the mistake. We have involved editing service from “Research Square” to assist with language corrections. We really hope that the flow and language level have been substantially improved.

Changes made:

We changed “The reversibility phase transition between the PV and BM phases of LSMO induced nonlinear relaxation after the removal of the stimulation from the gate.” to “The reversible transition between the PV and BM phases of LSMO induced nonlinear relaxation after the removal of the stimulation from the gate.” [paragraph 1, Page 15]

Moreover, we have involved editing service from “Research Square” to assist with language corrections, and revised the manuscript thoroughly.

REVIEWERS' COMMENTS

Reviewer #2 (Remarks to the Author):

In their revised manuscript and response, the authors have comprehensively addressed my initial comments and concerns. For this reason I enthusiastically recommend publication.

Reviewer #3 (Remarks to the Author):

The authors have addressed my concerns in their response and revised manuscript. I recommended it for publication as is.

Reviewer #2 (Remarks to the Author):

In their revised manuscript and response, the authors have comprehensively addressed my initial comments and concerns. For this reason I enthusiastically recommend publication.

Response:

We are grateful to the reviewer for the recommendation of this work.

Reviewer #3 (Remarks to the Author):

The authors have addressed my concerns in their response and revised manuscript. I recommended it for publication as is.

Response:

We are grateful to the reviewer for the recommendation of this work.